

# Quantifying horizontal and vertical tracer mass fluxes of a daytime valley boundary layer

Daniel Leukauf[1], Alexander Gohm[1], and Mathias W. Rotach[1]

[1]University of Innsbruck, Institute of Atmospheric and Cryospheric Sciences, Innrain 52f, A6020-Innsbruck, Austria

*Correspondence to:* Daniel Leukauf (Daniel.Leukauf@uibk.ac.at)

**Abstract.** The transport and mixing of pollution during the daytime evolution of a valley boundary layer is studied in an idealized way. The goal is to quantify horizontal and vertical tracer mass fluxes between four different valley volumes: the mixed layer at the valley floor, the slope wind layer, the stable core and the free atmosphere above the valley. For this purpose, large eddy simulations are conducted with the Weather Research and Forecasting (WRF) model for a quasi-two-dimensional valley.

The valley geometry consists of two slopes with constant slope angle, which rise to a crest height of 1500 m, and a 4 km wide flat valley floor in between. The valley is 20 km long and homogeneous in along-valley direction. Hence, only slope winds but no valley winds can evolve above this terrain. Various experiments are conducted for different surface heating and static stabilities. The surface sensible heat flux is horizontally homogeneous and prescribed by a sine function with amplitudes of $A_{\mathrm{shf}} =$ 62.5, 125, 250 and 375 W m$^{-2}$ respectively. The initial sounding is characterized by an atmosphere at rest and by a constant

Brunt-Väisälä frequency which is varied between $N = 0.006$ and $0.02$ s$^{-1}$. A passive tracer is released with an arbitrary but constant rate at the valley floor and resulting tracer fluxes are evaluated between the aforementioned volumes.

As a result of the surface heating, a mixed layer establishes in the lower part of the valley with a stable layer on top — the so-called stable core. The wind speed and, hence, the mass flux within the slope wind layer decreases with height due to the

vertically increasing stability of the environment near the center of the valley. Due to mass continuity, this leads to a partial redirection of the flow from the slope wind layer towards the valley center and the formation of a horizontal intrusion above the mixed layer. This intrusion is associated with a transport of tracer mass from the slope wind layer towards the valley center. For $A_{\mathrm{shf}} = 250$ W m$^{-2}$ and $N = 0.018$ s$^{-1}$, about 45 % of the tracer mass released at the valley floor is transported by this circulation to the stable core. This number rapidly decreases with decreasing stability and increasing forcing since the breakup

of the stable layer and, hence, the complete neutralization of the stratification in the valley is reached earlier. Between 60 and 85 % of the total tracer mass enters the slope wind layer at its lower end, but only a fraction is exported at crest height for high stabilities and a weak forcing. The total export of tracer mass drops from 80 % for the weakest stability and the strongest forcing to about 5 % for the strongest stability and the weakest forcing. The effects of initial stability and forcing can be combined to a single parameter, the breakup parameter $B$. An analytical function is presented that describes the exponential

decrease of the percentage of exported tracer mass with increasing $B$.

Keywords: slope winds, pollutant transport, vertical exchange, large eddy simulation



# 1 Introduction

It is well known that slope winds provide a mechanism for the vertical exchange of quantities like heat, moisture and pollutants between a valley atmosphere and the free atmosphere aloft (e.g., Reuten et al., 2007; Schmidli and Rotunno, 2010; Lehner and Gohm, 2010; Schmidli, 2013; Rotach et al., 2015). This exchange can be large during fair-weather summer days and the export of up to three times the valley air mass has been reported (Henne et al., 2004). However, during stable and low-forcing conditions (i.e, high buoyancy frequency and weak surface sensible heat flux), the vertical venting is limited and daily pollution limits are frequently exceeded (e.g., Chazette et al., 2005; Suppan et al., 2007; de Franceschi and Zardi, 2009; Silcox et al., 2011; Chemel et al., 2016). Studies have shown that the flow within the slope wind layer may be redirected horizontally leading to a partial recirculation below peak height (e.g., Vergeiner and Dreiseitl, 1987; Reuten et al., 2007; Princevac and Fernando, 2008; Gohm et al., 2009; Lehner and Gohm, 2010). This mechanism is sometimes called a pollution trap (Rendón et al., 2014) and may profoundly limit the vertical exchange with the free atmosphere. Stable conditions which lead to such recirculations are common in Alpine valleys throughout the whole year, but especially during fall and winter, giving rise to the question as to how effective this exchange mechanism really is. Hence, this study aims at quantifying the vertical and horizontal fluxes of pollutants for different atmospheric stabilities and different forcing strengths.

In the early work of Vergeiner and Dreiseitl (1987) on slope and valley winds, recirculations within the valley atmosphere are already mentioned. According to their work, horizontal motions appear at transition zones near elevated temperature inversions where the depth of the slope wind layer and/or the wind speed is reduced. As a result, the mass flux in the slope wind layer is reduced. Due to mass continuity, a fraction of the slope flow is recirculated below the inversion towards the valley center. Air from the valley center is incorporated into the slope wind layer above the inversion as the along-slope mass flux increases again. More recently Princevac and Fernando (2008) proposed a different mechanism for the formation of this type of flow splitting below the inversion. According to this study, a flow ascending along the slope leads to convergence if the slope flow at higher levels is either slower or down-slope directed. Due to the mixing within the slope wind layer, air parcels at its top are negatively buoyant with respect to the air in the valley core and either propagate more slowly upwards or even downwards. This leads to a partial redirection of the flow towards the center of the valley at mid-valley depth which they call horizontal intrusion (HI). This HI is a persistent flow feature, located at the transition zone. However, turbulence is generated as the HI penetrates into the stable core. Hence, it is a part of the mean flow field but has a turbulence component as well. Depending on slope angle, forcing and stability, this intrusion may play a role in the erosion of the cold pool in the valley and, hence, the breakup of the valley inversion. In general, HIs appear more likely in regions of changing slope angle, surface forcing or stability (Reuten et al., 2007). In addition to this HI, transport of heat and pollution out of the slope wind layer occurs at essentially all heights due to turbulent detrainment at the top of the slope wind layer.

In nature, rather rapid changes of stability or surface conditions with height are the norm rather than the exception. For example, elevated inversions have been found in an Alpine valley in 84 % of the time during a 18 days period in January (Emeis et al., 2007). When favourable conditions are met and pollutants are present at the valley bottom, HIs may create banners of polluted air. Such a case has been studied in the Inn Valley using airborne in-situ observations and a backscatter lidar





(Harnisch and Gohm, 2009; Gohm et al., 2009; Schnitzhofer et al., 2009). Three distinguishable banners with elevated aerosol concentrations were found that originated from the sun-lit slope of the valley and appeared to be formed by an interaction of the slope wind with elevated inversions. An idealized modelling study based on this case reinforces this conclusion (Lehner and Gohm, 2010). The presence of elevated layers with increased pollution is not only confined to valleys. Lu and Turco

(1994, 1995) have studied the transport of pollutants in the coastal environment of the Los Angeles Basin and the Santa Ana Mountains using an atmospheric model. The presence of the sea breeze interacting both with the convective boundary layer and the slope wind layer led to a complex interactions and the authors identified four different processes responsible for pollution layers. One mechanism is again the intrusion of aerosol-rich air from the mountain slope into an inversion layer.

The intrusion of aerosol and other pollutants into the inversion layer has implications for the definition and determination

of the boundary layer height over complex terrain. The convective boundary layer (CBL) or the mixed layer (ML) is usually defined over flat terrain as a layer with neutral stratification topped by an inversion (Sullivan et al., 1998). However, other definitions use the distribution of pollution and define the ML as the layer above ground, in which pollutants released at the surface are mixed within about an hour (Seibert et al., 2000). Since pollutants are also transported vertically by slope winds, and horizontally by detrainment from the slope wind layer and by HIs, this definition is ambiguous for a valley. If multiple aerosol

layer exist within a valley, the mixed layer height according to this definition may be unclear (De Wekker and Kossmann, 2015).

Interactions of the plain-to-mountain wind with valley winds and slope winds have been shown to be important for exchange processes within a valley atmosphere as well. Pollutants emitted above a plain adjacent to a valley are advected into the valley by the up-valley wind, transported to above crest height by slope winds, and are advected towards the plain again. This way,

pollutants can reach heights which they could not reach by only turbulent vertical transport above flat terrain (Wagner et al., 2015a). Lang et al. (2015) have shown that another venting mechanism is linked to valleys embedded in a mountain ridge. These embedded valleys develop a slope wind system of their own which interacts with the plain-to-mountain flow. The slope wind can be strong enough to prevent the plain-to-mountain flow from advecting pollution further towards the main peak of the mountain range which affects in turn the vertical distribution as well.

Inside a valley, even without an elevated inversion or a cold-air pool present at sun rise and with a homogeneous surface sensible heat flux, at least one HI is likely to develop during the evolution of the valley boundary layer. This is due to the development of a mixed layer with a capping inversion, which leads to a sharp transition from a neutral to a stable stratification and enables the development of an HI. For this reason a single HI formed in many previous simulations of the valley boundary layer (e.g., Wagner et al., 2014, 2015a, b; Leukauf et al., 2015; Schmidli and Rotunno, 2010; Schmidli, 2013; Serafin and

Zardi, 2010; Rendón et al., 2014). However, the role of intrusions in terms of mass transport have not been evaluated so far. In this study, we aim at quantifying the vertical fluxes of pollutants between the mixed layer, the slope wind layer, and the atmosphere above the valley as well as the horizontal fluxes out of the slope wind layer for a broad spectrum of atmospheric stabilities and amplitudes of surface sensible heat flux. The bulk export of pollutants is primarily relevant for the air quality inside the valley, but may also have an impact further away since polluted air is transported towards adjacent plains by alpine

pumping (Wagner et al., 2015a; Lang et al., 2015).



This study is organized as follows: In Sec. 2, the model setup and definitions of different volumes within the valley atmosphere are presented. Results of the simulations are presented in Sec. 3. The findings are discussed in Sec. 4 and conclusions are drawn in Sec. 5.

## 2 Methods

### 2.1 Model Setup

We are using the Weather Research and Forecasting model (WRF-ARW), version 3.4. This model has already been applied successfully for simulations of thermally driven winds in a previous study by Leukauf et al. (2015) and by other authors both with a kilometer-scale resolution and for large eddy simulations (LES) (Catalano and Cenedese, 2010; Wagner et al., 2014, 2015a, b).

The model configuration is similar to the one used in Leukauf et al. (2015), with only a few differences regarding terrain geometry, domain size, grid resolution, forcing and initial sounding. In this study we focus on the impact of stability and forcing strength and have chosen a model topography as simple as possible to avoid flow features introduced by terrain inhomogeneities such as a changing slope angle. It consists of a 4 km broad valley bottom, flanked by two slopes of constant slope angle rising over a width of 6 km to the crest at 1500 m. The crest on each side is characterized by a 500-m wide plateau. The topography is displayed in Fig. 1. In the horizontal, the domain consists of $340 \times 200$ grid points with a mesh size of 50 m which leads to a 17-km broad and 10-km long valley. The $x$-axis is cross-valley oriented and the $y$-axis is parallel to the valley. We are using a vertically stretched grid with 119 levels. At the valley bottom, the lowest full model level lies at 12 m and the second one at 25 m above the surface, leading to a minimum vertical grid spacing of $\Delta z = 13$ m. The model level distance increases vertically to a maximum of 75 m and the topmost model level is located at 8 km above mean sea level (amsl). The upper 2 km of the domain are occupied by a Rayleigh damping layer with a damping coefficient of 0.003 s$^{-1}$. The valley floor is at 0 m amsl and periodic boundary conditions are used both in $x$- and $y$-direction.

The surface sensible heat flux is prescribed using a sine-function with amplitudes of 62.5, 125, 250 and 375 W m$^{-2}$ and a period of 24 hours. The reference amplitude is 125 W m$^{-2}$ and the other values are one half, twice and three times the reference value, respectively. The model is initialized at 6 local time (LT) and run for 12 hours, hence, only daytime conditions are considered. Neither a boundary layer parametrization nor a land surface scheme is required. A 1.5-order three-dimensional turbulence kinetic energy closure (Deardorff, 1980) is used for the parametrization of sub-grid scale turbulence and a Monin-Obukhov similarity scheme (Monin and Obukhov, 1954) is applied at the surface.

The model is initialized with idealized soundings characterized by a constant Brunt-Väisälä frequency $N$. For the reference forcing of $A_{\mathrm{shf}} = 125$ W m$^{-2}$, $N$ is varied between 0.006 and 0.020 $^{-1}$ with an increment of 0.002 $^{-1}$. For the other cases, only four simulations are carried out for each $A_{\mathrm{shf}}$ with $N = 0.006, 0.010, 0.014$ and 0.018 $^{-1}$. In all cases the surface temperature is $T_0 = 280$ K, the surface pressure $p_0 = 1000$ hPa at $z = 0$ m amsl and the relative humidity is 40 % at all levels. The atmosphere is initially at rest. In order to trigger convection, a randomly distributed perturbation of 0.5 K is added to the base state potential temperature at the lowest five model levels. The simulations are labeled with acronyms such as S1N10, where 'S' stands for





'sine forcing', the following number is the factor with which the reference forcing amplitude of 125 W m$^{-2}$ is multiplied, $N$
refers to the Brunt-Väisälä frequency and the number afterwards is its value in $10^{-3}$ s$^{-1}$.

The passive tracer, that is used to measure the horizontal and vertical exchange rates, is released at the lowest model level
between $x = $ -2 km and +2 km, i.e., at the valley bottom and along the whole length of the valley. The emission rate is set to
one unit per time step, an arbitrary but constant value. The emission starts immediately at the beginning of the simulation, i.e.,
at 6 LT.

## 2.2 Averaging Methods

The procedures used to calculate vertical and horizontal fluxes follow the averaging approach of Schmidli (2013) and are the
same as in Leukauf et al. (2015), but applied to tracer mass fluxes.

A variable $\tilde{\phi}$ consists of a component resolved by the model $\bar{\phi}$ and a subgrid-scale component $\phi'$. Averaging the model grid
variable $\bar{\phi}$, gives the mean component $\langle\bar{\phi}\rangle$, and subtracting the mean from $\bar{\phi}$ obtains the resolved turbulence component (i.e.,
$\phi'' = \bar{\phi} - \langle\bar{\phi}\rangle$). In total, $\tilde{\phi}$ is decomposed into mean, resolved turbulence and subgrid-scale turbulence component:

$$\tilde{\phi} = \langle\bar{\phi}\rangle + \phi'' + \phi'. \tag{1}$$

The averaging $<^-\!>$ is hereby defined as an average along the valley and over 41 minutes. In the WRF model, the tracer is
defined by a non-dimensional mixing ratio $\bar{r}$. Hence the tracer density is $\bar{\rho}_{tr} = \bar{r}\bar{\rho}$, where $\bar{\rho}$ is the air density. The total vertical
tracer mass flux in $z$-direction can then be decomposed into:

$$\underbrace{\langle\,\overline{\tilde{w}\,\tilde{\rho}_{tr}}\,\rangle}_{\text{TOT}} = \underbrace{\langle\bar{w}\rangle\,\langle\bar{\rho}_{tr}\rangle}_{\text{MEA}} + \underbrace{\langle\overline{w''\rho_{tr}''}\rangle}_{\text{RES}} + \underbrace{\langle\,\overline{w'\rho_{tr}'}\,\rangle}_{\text{SGS}}. \tag{2}$$

The same decomposition can be applied to the horizontal tracer mass flux $\langle\,\overline{\tilde{u}\,\tilde{\rho}_{tr}}\,\rangle$.

### 2.3 Volumes and bulk fluxes

As long as the valley is characterized by at least one stably stratified layer and anabatic slope winds exist, the valley volume
can be divided into three different sub-volumes (see Fig. 1).

The first volume $V_1$ is a mixed layer in the lower part of the valley, i.e., the convective boundary layer (CBL) up to the
height $h_{\text{cbl}}$. The second volume $V_2$ is the *sum* of the two slope wind layers on the two valley sides. $V_2$ is defined by the depth
of the slope wind layer, $d_{\text{swl}}$, the upper boundary at crest height, $h_c$ and the lower boundary at the CBL height averaged over
the valley floor $\bar{h}_{\text{cbl}}$. One could argue that the slope wind layer also exists over the slopes in $V_1$. However, it is hard to separate
the slope wind layer form the mixed layer in this region, so that the chosen definition is more useful. The rest of the valley
volume is occupied by the stable core and is denoted by $V_3$. Finally, above the valley is the free atmosphere which acts as a
forth volume $V_4$. Notice that $\bar{h}_{\text{cbl}} = \bar{h}_{\text{cbl}}(t)$ and $d_{\text{swl}} = d_{\text{swl}}(x,t)$. Hence, the volumes $V_1$, $V_2$ and $V_3$ are time dependent and
$V_4$ is assumed to be time invariant. At sunrise, when neither a slope wind layer nor a mixed layer exists, the whole valley is
occupied by $V_3$. We call this initial state *regime 0*. Figure 1 illustrates the subsequent *regime 1* with well-defined $V_1$, $V_2$ and





$V_3$. When the stable core is completely eroded, i.e., the so-called temperature inversion breakup is reached, volumes $V_2$ and $V_3$
become zero and the valley is completely occupied by $V_1$. We denote this state by *regime 2*.

Tracer fluxes integrated over the interface between two volumes are denoted by $F_{ij}$, with the two indices $i$ and $j$ referring
to the two volumes between which the flux is defined (see Fig. 1). A flux directed from $V_i$ to $V_j$ is counted positive. All
fluxes are defined to be normal to their respective boundary. Since the valley geometry and the corresponding slope flow
circulation is symmetric with respect to the valley center, we treat the two slope wind layers as a single volume $V_2$. Hence, $F_{23}$
represents the sum of the two bulk fluxes from the two slope winds layers into $V_3$ or vice versa. Finally, the spatial integral of
the prescribed constant surface tracer mass flux at the bottom of the valley is denoted as $F_{01}$. It is useful to normalize fluxes
between volumes with the flux at the surface, i.e., $f_{ij} = F_{ij}/F_{01}$. Likewise, the integral of the flux over the duration of the
simulation (12 h) reveals the mass passing the border between two volumes, which is denoted by $M_{ij}$. Correspondingly, the
total tracer mass released at the surface is $M_{01}$ and the normalized tracer mass crossing the interface between volumes $i$ and $j$
is $m_{ij} = M_{ij}/M_{01}$. Due to the averaging operations which are performed along the $y$-axis and over 41 minutes, the first point
in time where full averages are available is 0645 LT. The integral over time is based on model output stored every five minutes.
The tracer mass inside a volume at a given point in time is denoted by $M_i^t$ and the total tracer mass released until this time as
$M_{01}^t$. Hence $M_{01}^t(t = 18\text{ LT}) = M_{01}$.

Vertical tracer fluxes are calculated by integrating the total tracer mass flux $\langle \overline{\tilde{w}\,\tilde{\rho_{tr}}} \rangle$ (see Eq. 2) along the border between the
respective volumes. It is important to keep in mind that the length and position of the borders between volumes change in time
as the boundary layer evolves. Hence, the volumes $V_1$ to $V_3$ are not constant in time, which leads to tracer mass tendencies due
to the change of the respective volume size. Since the mixing layer grows during the day, tracer mass is re-incorporated into
$V_1$ by this process while $V_2$ and $V_3$ loose some tracer mass this way. This is a relatively slow, but relevant process. This aspect
has to be considered when comparing budgets and fluxes between different points in time. Since the total tracer mass flux is
calculated on full model levels, an interpolation is necessary to calculate correct fluxes between two grid cells. For vertical
fluxes, this interpolation must be done only between two grid cells located above each other, while for horizontal fluxes, the
change in height due to terrain following coordinates has to be taken into account. The interpolation is error prone when applied
to a region where fluxes have strong gradients. This is especially the case for $F_{23}$ for which the intersection between $V_2$ and $V_3$
passes right through an area with strong horizontal flux gradients.

For this reason the flux $F_{23}$ is calculated indirectly as a residuum from the tracer mass tendency and the vertical fluxes. This
can be done by either relating to $V_2$ or to $V_3$. The equation for $V_2$ is:

$$F_{23} = \underbrace{L_y \int_{L_{12}} \langle \overline{\tilde{w}\,\tilde{\rho_{tr}}} \rangle \, dx}_{F_{12}} - \underbrace{L_y \int_{L_{24}} \langle \overline{\tilde{w}\,\tilde{\rho_{tr}}} \rangle \, dx}_{F_{24}} - L_y \int_{A_2} \frac{\partial \langle \bar{\rho}_{tr} \rangle}{\partial t} dA, \tag{3}$$



and similarly for $V_3$:

$$F_{23} = \underbrace{L_y \int_{L_{13}} \langle \overline{\tilde{w}\, \tilde{\rho_{tr}}} \rangle\, dx}_{F_{13}} \underbrace{- L_y \int_{L_{34}} \langle \overline{\tilde{w}\, \tilde{\rho_{tr}}} \rangle\, dx}_{F_{34}} - L_y \int_{A_3} \frac{\partial \langle \bar{\rho}_{tr} \rangle}{\partial t}\, dA, \tag{4}$$

Here, $A_i$ is the vertical side face of $V_i$ on the $x$-$z$ plane and $L_{ij}$ the borderline between $V_i$ and $V_j$ in $x$-direction. The local tendencies in Eq. 3 and 4 are integrated only over cross sections on the $x$-$z$ plane since they are already averaged in $y$-direction. However, the multiplication with $L_y$, the length of the valley, yields terms representative for the whole volume. The vertical fluxes on the right side of Eq. 3 and 4 are validated by comparing the volume integrated tracer mass tendencies in $V_1$ and $V_4$ with the sum of fluxes in and out of these volumes and good agreement is found. The calculation of $F_{23}$ is done for $V_2$

and $V_3$ individually according to Eq. 3 and 4, respectively. The agreement between these two independent results for $F_{23}$ is satisfying. The difference is largest in the first two hours with about 20 % relative to $F_{01}$, but decreases afterwards to nearly zero. Henceforth, we use an average based on the two values from Eq. 3 and 4 to minimize the effects of errors especially in the early stage of the simulation.

### 2.4    Definitions of boundary layer and slope wind layer

In order to quantify tracer fluxes between the volumes described above, working definitions of convective boundary layer height $h_{cbl}$ and slope wind layer depth $d_{swl}$. Here, $d_{swl}$ is the vertical distance between terrain surface $h_s$ and upper boundary of the slope wind layer, whereas the slope wind layer height $h_{swl}$ is its absolute height, i.e., $h_{swl} = h_s + d_{swl}$. Defining the boundary layer height over complex terrain is not straight forward since $h_{cbl}$ may strongly depend on terrain and synoptic conditions (De Wekker and Kossmann, 2015). Herein, we calculate the average convective boundary layer height $\bar{h}_{cbl}$ above the flat valley

floor based on model profiles averaged between $x = -2$ km and $x = +2$ km using a criterion described below. The daytime slope wind layer (SWL) has similar characteristics to a convective boundary layer (CBL) over a flat surface such as the valley floor. Hence, for determining $h_{swl}$ we use the same criterion as for $h_{cbl}$. Consequently, the CBL height over the flat part of the valley will smoothly change into the slope wind layer height at $x = \pm 2$ km.

In principle, various definitions of $h_{cbl}$ and $h_{swl}$ are conceivable. First, the convective boundary layer height can be defined

as the lowermost level where the vertical gradient of the mean potential temperature $< \bar{\theta} >$ exceeds the value of 0.001 K m$^{-1}$ when moving upward from the surface. This definition is similar to the one of Catalano and Moeng (2010), however, without additional constraint on the heat flux. We shall refer to this definition as the $\theta$-gradient-criterion. The classical definition refers to the turbulent nature of the CBL and defines the CBL height as the level where the total vertical heat flux has its negative minimum (Sullivan et al., 1998). This definition can lead to an artificial step-like structure of the CBL height in space and

time when the heat flux minimum is not well defined at a distinct level but is rather spread over a certain altitude range. For this reason, we slightly modify this criterion and determine the CBL height at the level $i$ at which the decrease of the vertical heat flux between $i$ and $i+1$ is not more than 5 %. This criterion shall be called heat-flux-criterion. As mentioned above, both criteria could be used not only to determine the CBL above the valley floor, but also to identify the slope wind layer depth. Yet





another way to define $h_{swl}$ is to separate the upward flow inside the SWL from the subsidence in the center of the valley. Here,
the SWL is the layer above the slope where the mean vertical wind component $<\bar{w}>$ is positive. This definition is implicitly
used, e.g., in the analytical model of Vergeiner and Dreiseitl (1987) since it characterizes the upward mass flux that determines
export processes. Is is only meaningful above the slopes. We shall call this definition the $w$-criterion.

Boundary layer heights and SWL depths based on these three definitions are shown for simulation S1N10 in Fig. 2. In the
morning until noon, the $\theta$-criterion gives lower values of $h_{cbl}$ above the valley floor than the heat-flux-criterion. Over the slopes,
the two definitions reveal very similar $d_{swl}$, with the one from the $\theta$-gradient-criterion again yielding the lowest boundary layer
height. However, as the stable core is eroded, the CBL height according to the $\theta$-gradient-criterion reaches much greater
heights than the one according to the heat-flux-criterion since it is sensitive to the threshold value of $0.001\ \mathrm{K\ m^{-1}}$. This feature
occurs quite suddenly as soon as the vertical gradient of the potential temperature falls below this threshold. Choosing a lower
threshold would only delay but not prevent this behaviour. For this reason, the $\theta$-gradient-criterion is not suitable for weak
initial stabilities and later in the afternoon. The $w$-criterion gives SWL depths, which are very similar to those generated with
the heat-flux-criterion within the slope wind layer.

Given the sensitivity to small temperature gradients of the $\theta$-gradient-criterion and the failure of the $w$-criterion to yield
a suitable CBL height in the center of the valley, we decided to use the *heat-flux-criterion* in the remainder of the paper for
both the CBL and SWL height which in turn define the boundaries of volumes $V_1$, $V_2$ and $V_3$. Here, $\bar{h}_{cbl}$ is used as the upper
(lower) boundary of $V_1$ ($V_2$ and $V_3$) while $h_{swl}$ serves as the lateral boundary between $V_2$ and $V_3$ (see Fig. 1). A spatial average
using a Hann-window over 2 km is applied to $d_{swl}$ before calculating $h_{swl}$ to smooth wiggles which would be unsuitable for the
definition of $V_2$ and its border to $V_3$.

# 3   Results

## 3.1   General flow and tracer distribution

The general flow and tracer distribution are described in this section for the S1N10 simulation, which serves as a reference.
The evolution the other simulations is discussed briefly at the end of this section.

At the start of the simulation (6 LT) the whole atmosphere is stably stratified and characterized by a constant buoyancy
frequency (*regime 0*). With the onset of the surface-layer heating in the morning, a shallow mixed layer begins to form in the
center of the valley and a slope wind layer (SWL) develops (*regime 1*). This layer consists of thermals that are advected by
the slope winds and rise along the slope (Fig. 2a). The turbulent nature of the SWL leads to mixing at the SWL top, a process
which leads to its growth, but also to exchange of air with the stable core. Averaging in time and along the $y$-axis reveals not
only a mean up-slope flow and the recirculation above the valley, but also the formation of a horizontal intrusion just above the
CBL height, next to the lower end of the slope wind layer (Fig. 2c). The slope wind layer is deeper at lower levels near $\bar{h}_{cbl}$ and
decreases with increasing height. The wind speed close to the surface has a comparable magnitude along the slope. However,
as $d_{swl}$ decreases with height, the mass flux inside the SWL decreases as well, which leads to along-slope convergence. Due
to mass continuity, this convergence leads to an export of air from the SWL to the valley center and, hence, the formation of a





horizontal intrusion (HI). The intrusion is stronger at 12 LT since the reduction of $d_{swl}$ with height is much stronger at 12 LT than at 09 LT (Fig. 2c,d).

Tracer released at the valley bottom is distributed homogeneously within the mixed layer and is also advected upwards by slope winds (Fig. 2e). In simulation S1N10, the tracer reaches the crest height about 2.5 hours after the simulation has started. Tracer mass export from the SWL to the center of the valley is strongest in the lower part of the stable core where the mass flux reduction in the SWL is strongest too. It is noteworthy, that no well-defined horizontal banners of tracer develop within the valley as observed, e.g., for example in the Inn Valley (Harnisch and Gohm, 2009), but rather a deep layer of tracer without clear separation from the CBL underneath. This is due to the vertically constant initial stability $N$ we have chosen for this idealized approach. An elevated inversion, much higher than $h_{cbl}$, would promote the formation of tracer banners. The HI, being an phenomenon caused by the interaction of the slope wind layer with the stable core, forms as soon as the SWL develops and persists as long as the stable core and the upslope flow exist. This persistence allows the HI to redistribute considerable amounts of tracer mass despite the relative low mean horizontal wind speed, which ranges between 0.5 and 1.0 m s$^{-1}$. In addition to the mean advection by the HI, tracer is transported to the stable core by turbulent mixing at the top of the CBL above the valley floor as well as above the slopes.

As the boundary layer evolves, the slope wind layer grows in depth and the wind speed within the HI increases (Fig. 2c,d). At 12 LT, tracer mass, which has reached the atmosphere above the valley, is distributed in an almost neutral layer which has developed due to a second and stronger circulation above the valley. This circulation is ultimately responsible for reimporting tracer mass through the valley top into the stable core (Fig. 2f). With continuous heating, the valley atmosphere would eventually become completely neutral and reach *regime 2*. However, the reference case S1N10 has a forcing strength just slightly too weak for the breakup, but cases with a stronger forcing or a weaker stratification do reach this regime. If this is the case, the mixed layer inside and the neutral layer above the valley merge, creating a deeper mixed layer which extends beyond crest height. Within this layer, turbulent mixing is very effective and with a constant source at the bottom, this leads to large vertical tracer fluxes at crest height.

The tracer fluxes responsible for the redistribution of tracer mass are shown in Fig. 3 for S1N10 at 12 LT. Tracer is transported along the slopes primarily by mean horizontal and vertical fluxes (Fig. 3a,b). At the border of $V_2$, defined by the SWL depth, strong horizontal fluxes carry tracer mass towards the valley center while weaker downward fluxes indicate a partial recirculation of tracer mass into the mixed layer. Horizontal transport due to the circulation above the valley is also indicated by a strong mean tracer flux. Overall, the horizontally resolved turbulent flux is one order of magnitude smaller than the corresponding mean flux (Fig. 3c). The vertical resolved turbulent flux is positive in the ML and the SWL and strongest in magnitude in the center of the valley (Fig. 3d). Over the slopes, the vertical resolved turbulent flux is, again, by about one order of magnitude smaller than the mean vertical tracer flux, but extends further up into the stable core (Fig. 3d).

Simulations with a stronger (weaker) forcing or a weaker (stronger) stability than the reference case S1N10 exhibit a similar evolution, but the inversion breakup, i.e., regime 2, is reached much earlier (later). For example, the breakup is reached after about 4 hours for S1N06, thus allowing for a larger amount of tracer export from the valley atmosphere, whereas for simulations S1N12 to S1N20 the breakup is never reached, which restricts vertical venting considerably. In terms of structure, tracer fluxes



have a similar pattern in all simulations as long the breakup is not reached. However, for a stronger stratification, the slope wind layer is shallower and the mixed layer grows slower. The tracer flux at the top of the SWL is weaker and the recirculation within the valley is more pronounced. Only a single HI develops in all cases.

## 3.2 Bulk fluxes between boundaries

Figure 4 shows the normalized fluxes $f_{ij}$ as a function of time for the simulations S1N06 to S1N12 and S1N16. In the early morning, both fluxes out of $V_1$, are positive were $f_{13}$ is larger than $f_{12}$ and has a strength of about 25 to 50 %. The flux from $V_1$ into the SWL is zero in the beginning but grows steadily as the slope winds intensify with increasing surface sensible heat flux. The flux between the stable core towards the SLW is negative, which means that it is directed towards the SWL and advects some of the tracer brought into $V_3$ by $f_{13}$. It has values in the order of 12.5 % but decreases and is zero at about 0900 LT. The flux $f_{23}$ is negative in the early morning hours if the convective boundary layer is still very shallow ($< 500$ m) and air enters the SWL at such low levels not only vertically but also horizontally. Until 0900 LT, the evolution of the fluxes in all S1 simulations is very similar.

At about 0830 LT, the tracer reaches crest height and is exported out of the valley. Hence, flux $f_{24}$ increases, but its maximum strength depends strongly on the initial stratification. In case of the weakest stratification (N06) this flux reaches about 50 % before the breakup at about 1030 LT. In this case, not only $f_{24}$ is positive, but also $f_{34}$ which indicates that even in the center of the valley more tracer is exported than reimported at crest height. The tracer mass export out of the SWL increases slower for simulations with a stronger initial stratification and reaches much smaller peak values. For example, in S1N08, $f_{24}$ has a peak value of 100 % at noon while for S1N16 this flux has a maximum of only about 25 %. If the inflow of tracer mass at the lower boundary of $V_2$ is greater than the outflow at crest height, tracer mass accumulates in this volume. Eventually $f_{23}$ reverses its sign as tracer mass is advected from the SWL to $V_3$. This flux is weak for S1N08 and decreases again once $f_{24}$ has similar values as $f_{12}$. For strong initial stratifications (i.e., N12 and stronger), when $f_{24}$ is always smaller than $f_{12}$ it is large with peak values ranging between 60 and 80 %. For these simulations, $f_{23}$ is the main source of tracer mass for the stable core.

In the afternoon, simulations differ greatly from each other depending on the initial stability. While simulations with a weak stability reach the breakup either before noon (S1N06, Fig. 4a) or in the early afternoon (S1N08, Fig. 4c), this does not happen for stronger initial stabilities (Fig. 4d,e,f). In regime 2, the only remaining flux is $f_{14}$ which climbs to values of about 100 %. For the weakest stability, it oscillates in the afternoon between 60 and 125 %. The reason for these strong oscillations are large convective cells which develop in the boundary layer and redistribute tracer mass vertically. If the valley would be longer, the averaging area would be larger and those oscillations would presumably be much smaller. For S1N08, the thermals in the center of the valley are weaker and $f_{13}$ is practically zero at breakup time. This flux is even slightly negative before the breakup when tracer is reimported from $V_4$ again. Hence, during regime 1, the whole export of tracer mass takes place at the interface between $V_2$ and $V_4$, which leads to a smooth transition from $f_{24}$ to $f_{14}$.

For the case S1N10 the forcing is just too weak to remove the given inversion completely which leads to large maximum export of tracer mass but it grows more slowly compared to simulations with a weaker initial stratification. Due to this slower increase, more tracer mass accumulates before noon in $V_1$ leading to peak values for $f_{24}$ of 125 %. Since the inversion is never





completely eroded, thermals at the center of the valley do not reach out of the valley. Instead, tracer mass is reimported at crest height by subsidence. The tracer mass cannot leave the vicinity of the valley due to the periodic boundary conditions for which reason the reimport of tracer mass is very larger and reaches a value of -75 % in the late afternoon. Hence, the net export of tracer mass peaks at about 45 % at 1400 LT. Some of this reimported tracer is again reintroduced into the SWL below peak height leading to a negative $f_{23}$ (Fig. 4d). For even stronger initial stabilities, the subsidence is smaller and the amount of tracer mass exported is greatly reduced. Hence, hardly any tracer mass is reimported into the valley volume. The tracer mass flux is instead dominated by the inflow into the SWL and the outflow into $V_3$. This outflow, $f_{23}$, has an amplitude up to three times as large as the outflow from the SWL to the free atmosphere $f_{24}$. However, the tracer mass advected into the stable core remains only partially in this volume since $f_{13}$ turns negative in the afternoon and reaches values of up to -50 %. Hence, tracer mass is advected back into $V_1$, a flux pattern which nicely illustrates the circular motion of tracer inside the 'pollution trap'. The efficiency of this circulation increases with increasing static stability $N$. For the strongest stability shown in Fig. 4f, the peak value of net tracer export to the free atmosphere ($f_{24} + f_{34}$) in the early afternoon is only about 25 %. The fluxes in the simulations with the strongest stabilities, S1N14 to S1N20, are very similar, for which reason only S1N16 is shown in Fig. 4f.

In the last half hour before sunset (1800 LT), $V_1$ shrinks quickly since the lack of heating from the surface leads to much lower convective boundary layer heights and SWL depths. The same happens to $V_2$ if this volume is still defined, i.e. when the breakup was never reached. Otherwise, the volumes $V_2$ and $V_3$ are re-established (Fig. 4a,c). This leads to sharp changes of associated fluxes. This sudden change of the volume sizes depends to some degree on the choice of the criterion used to define the volumes. If the $\theta$-gradient-criterion would be used instead of the heat-flux-criterion, the CBL-height defining the upper border of $V_1$ will suddenly drop as well, but this happens at sunset, and not half an hour before.

### 3.3 Total tracer mass transport

The total tracer mass $M_{ij}$ transported between $V_i$ and $V_j$ between 6 LT and 12 LT is defined as the integral of $F_{ij}$ over the course of the simulation. Notice that the volumes are not constant in time. Hence, the total mass transport does not allow for conclusions on the concentration of pollutants, but is a measure of the average strength of the respective fluxes. Figure 5 shows $M_{ij}$ normalized by the total mass released at the surface, $m_{ij} = M_{ij}/M_{01}$, for simulations S1N06 to S1N20. In this overall picture, it becomes obvious that the vertical transport out of $V_1$ is strong for all stabilities, since about 75% of the tracer mass is either transported into the SWL ($m_{12}$), directly into the stable core ($m_{13}$), or to the free atmosphere ($m_{14}$). The latter happens in case of low stabilities after the breakup of the stable stratification in the valley has been reached. Since $f_{13}$ is directed upwards in the morning and downwards in the afternoon, the direct net transport from $V_1$ to $V_3$ is relatively small. For the weakest stabilities (S1N06 and S1N08), $f_{13}$ is no longer defined after the breakup, so that the total mass transported from $V_1$ to $V_3$ ($m_{13}$) remains positive but, with some 20 %, relatively small. Once the tracer mass is in the SWL, it strongly depends on the atmospheric stability where it will be subsequently transported to. For weaker stability, the tracer mass is predominately exported out of the valley, i.e., large $m_{24}$ and/or $m_{14}$. In the case of N06, a total of about 70 % of the tracer mass released at the surface is exported. This number drops to about 15 to 20 % for N14 to N20, while the mass transport from the SWL to the stable core increases continuously with increasing stability and reaches values as large as 42 %. Before the breakup, tracer




mass is exported by the slope winds ($m_{24}$) and partially reimported by compensating subsidence near the center of the valley ($m_{34}$). In case of strong stability, hardly any tracer mass is reimported since there is little mass above the valley that could be reimported by subsidence. For weak stabilities, the fluxes associated with export and reimport are combined in $f_{14}$ once the breakup is reached. Since some tracer mass must be accumulated in $V_4$ to allow a significant reimport of tracer mass, $m_{34}$ is close to zero for S1N06 and S1N08. Only in the case of S1N10 a significant amount (25 %) is reimported via $m_{34}$ since this is a simulation with a stability weak enough to allow for a substantial export of tracer, but strong enough to avoid the breakup, so that $V_3$ exists even in the late afternoon.

The distribution of tracer within the valley as the fraction of tracer mass in each volume $V_i$ (i.e., $M_i^t/M_{01}^t$) is shown in Fig. 6 for simulations S1N06 to S1N20 for 17 LT. This point in time is chosen rather than 18 LT since the volumes $V_2$ and $V_3$ shrink or are re-established, respectively, as the surface sensible heat flux decreases sharply near the end of the simulation leading to sudden changes of the volume size. Hence, for the calculation of tracer concentrations, the situation at 17 LT is more representative for the situation close to sunset than the actual time of sunset. The tracer mass in $V_4$ (i.e., $M_4^t/M_{01}^t$), which corresponds to the total exported tracer mass, decreases in a non-linear fashion with increasing $N$. It drops from more than 60 % for S1N06 to only about 10 % for S1N20. The tracer mass fraction in the mixed layer $V_1$ is always between 40 and 55 %, whereas its volume is heavily dependent on the stability and drops sharply as the critical initial stability that allows for a complete erosion of the stable core is exceeded. Hence, a similar mass in a smaller volume leads to higher tracer concentration in the mixed layer for higher stabilities. For S1N12 to S1N20, the fraction of tracer mass in the stable core is approximately constant at 20 % whereas its volume increases with $N$. The volume of the slope wind layer accounts for only about 10 % of the valley volume and contains about 6 to 12 % of the tracer mass.

### 3.4 Dependence on the forcing amplitude

The time of the inversion breakup, and consequently whether it is reached at all before sunset, is an important parameter in describing the tracer mass fluxes between and concentrations within the various valley volumes. This time strongly depends on initial stability and forcing amplitude. The existence of a HI is dependent on the presence of a stable core, and the exchange towards the free atmosphere increases once the valley atmosphere is completely neutral. For simulations with different than the reference forcing (S1 corresponding to 125 W m$^{-2}$ amplitude) both the overall structure of the valley atmosphere and the magnitudes of tracer fluxes between the various volumes bear large similarities to the conditions in S1. Because of these similarities, we restrict the following analysis to the fluxes at the top and the bottom of the slope wind layer.

Figure 7a shows the evolution of $f_{12}$ for S0.5N10 to S3N10 and S1N06 to S1N18. The magnitude of the vertical flux into the slope wind layer and its increase in time primarily depends on the forcing strength. Different stabilities show a very similar evolution. This can be understood by considering the mass flux along the slope wind layer following the simple analytical slope-flow model of Vergeiner (Vergeiner, 1982; Vergeiner and Dreiseitl, 1987). According to this model, the mass flux $V\delta$ is:

$$V\delta = \frac{\frac{H}{\tan(\gamma)}(1-Q)}{c_p\rho\frac{d\Theta}{dz}},$$

(5)





where $V$ is the slope-parallel wind speed averaged over the SWL, $\delta$ the thickness of the SWL (i.e., the depth of the SWL measured normal to the slope), $(1-Q)H$ is the fraction of the surface sensible heat flux which heats the SWL, $\gamma$ is the slope angle, $\frac{d\Theta}{dz}$ is the potential temperature gradient in the center of the valley as a function of $z$ and the other symbols have their usual meaning. The stability at the lower end of the SWL is very similar in all simulations since this height corresponds to the boundary layer height. This explains the rather weak dependence on initial stratification. In contrast, a stronger forcing leads to a stronger tracer mass flux. The time of the breakup varies between 0745 LT (S3N06, not shown) and no breakup.

For the export of tracer mass out of the SWL into the free atmosphere above the valley, both stability and forcing are important factors. A stronger forcing results in a stronger up-slope flow and, hence, larger vertical flux $f_{24}$ (see Fig. 7b). Note that the decrease of tracer export is related to the decrease of forcing amplitude in a non-linear fashion. While S1N10 has a peak magnitude of about 125 % at 15 LT, S0.5N10 peaks at merely 25 %. A similar reduction of peak export $f_{24}$ also results if the stability is increased significantly beyond the critical value that enables the breakup (cf. S1N10 and S1N14). Interestingly, simulation S1N10 has the largest peak flux out of the SWL, even when compared to simulations with a stronger forcing or a weaker initial stability. The reason is the accumulation of tracer mass in $V_1$ during the day due to the fact that the breakup is just not reached. In combination with relatively strong slope winds, this leads to a peak export of 125 % once the tracer-rich air is transported through the SWL to the peak. For S2N10 and S3N10 this peak value is only about 100 %. The export out of the SWL for N1N06 peaks at about 60 % since the simulation reaches the breakup point and the export is henceforth part of $f_{14}$. It is obvious, that simulations with a strong inflow of tracer at the lower end of the SWL but a weak outflow at the top are characterized by a strong outflow of tracer mass into the stable core (not shown).

The dependence of the total tracer mass transport between $V_i$ and $V_j$ on the forcing amplitude $A_{\mathrm{shf}}$ is shown in Fig. 8 for N10 and N18. For the weaker stability, N10 (Fig. 8a), the total mass transport between SWL and stable core is zero, except for the weakest forcing S0.5. The import of tracer mass into the SWL is slightly higher than the export at crest height with the difference remaining inside the SWL resulting in a zero net transport to the stable core. This is not the case for the weakest forcing, in which the tracer inflow at the lower end of the SWL is about 60 % but the outflow only about 15 %. Consequently, tracer is redirected toward the stable core, with about 30 % and the rest, about 15%, remains in the SWL. The tracer transport from the mixed layer directly into the stable core ($m_{13}$) is about 10 to 15 % and only weakly depending on the forcing.

In case of the stronger stability, N18 (Fig. 8b), the inflow of tracer into the SWL is always greater than the outflow which leads to an horizontal transport towards the stable core on the order of 40 %. This flux increases with increasing forcing except for the strongest forcing (S3) where a much stronger vertical outflow at crest height leads to a weaker total flux between $V_2$ and $V_3$. There is also a recirculation from $V_4$ to $V_3$ and then again into $V_2$ close to crest height. This recirculation implies a negative *local* flux from $V_2$ to $V_3$ near crest level and, hence, reduces the bulk flux $m_{23}$. The direct transport of tracer between the mixed layer $V_1$ and the stable core $V_3$ is negative for this stability with values reaching -12 %. This is a sign of an effective recirculation at boundary layer height, i.e., part of the tracer that is exported from the mixed layer by slope winds is imported again into $V_1$ by turbulent mixing and the low level recirculation within the 'pollution trap'.





The total export of tracer mass ($m_{\text{tot}}$) increases almost linearly with the forcing for N18. For N10 this increase is stronger

for low forcing amplitudes but flattens for stronger forcing. Between 13 and 64 % of the tracer mass is exported for N10 but

only between 5 and 31 % is exported for N18.

### 3.5    Total export of tracer mass

It is clear from the previous analysis, that the breakup time is an important time scale for the evolution of the valley boundary

layer which depends on both the initial stability and the forcing amplitude. Hence, the venting of a valley strongly depends in

a non-linear way on both parameters, but the total export of tracer mass at crest height may be similar for certain combinations

of forcing and stability. A run with a strong forcing and a strong stability exports about as much tracer as a run with a weaker

forcing and a weaker stability (Fig. 9a). The question arises whether a common parameter can be found that combines the

effect of stability and forcing on the exchange of tracer. The answer will be given below.

After Whiteman and McKee (1982), the breakup time of a valley inversion primarily depends on $Q_{\text{req}}$, the energy required

to remove the inversion, evaluated at sunrise:

$$Q_{\text{req}} = L_y c_p \int\limits_0^{h_c} \rho(z) \left(\Theta_E - \Theta(z)\right) W(z) dz, \tag{6}$$

and $Q_{\text{prov}}^{\text{pre}}$, the energy provided by the surface sensible heat flux until the time of the breakup:

$$Q_{\text{prov}}^{\text{pre}} = \int\limits_{t_r}^{t_b} \int\limits_A H_s(t, x, y) dx \, dy \, dt. \tag{7}$$

Here, $H_s(t, x, y)$ is the sensible heat flux at the surface, $A$ is the surface area of the valley, $t_r$ and $t_b$ the times of sunrise and

breakup, respectively, $\rho$ is the air density, $L_y$ the length of the valley, $h_c$ the crest height, $\Theta_E = \Theta(z = h_c)$ and $W(z)$ is the

width of the valley as a function of $z$.

Combining Eq. 6 and 7 and assuming $Q_{\text{req}} \sim Q_{\text{prov}}^{\text{pre}}$ (see Leukauf et al. (2015) for the rational of this approximation) one

can see that a strong forcing and a weak initial stratification lead to an early breakup. However, not further conclusions for the

venting potential can be drawn from this calculation. Moreover, not only the breakup time but also the forcing strength after

the breakup is important since it will determine the strength of the vertical transport once the valley atmosphere is neutral.

It is therefore useful to calculate the total energy provided until sunset $t_s$:

$$Q_{\text{prov}} = \int\limits_{t_r}^{t_s} \int\limits_A H_s(t, x, y) dx \, dy \, dt, \tag{8}$$

The ratio of required and provided energy,

$$B = \frac{Q_{\text{req}}}{Q_{\text{prov}}}, \tag{9}$$





is a non-dimensional parameter. Values of $B \gtrsim 1$ describe conditions with strong stability and weak forcing that prevent an inversion breakup. In contrast, $B \lesssim 1$ implies weak stability or strong forcing that enables an inversion breakup. Hence, the smaller $B$, the earlier the earlier occurs the breakup. For this reason we call $B$ the *breakup parameter*. As a first approximation one can say that for $B = 1$, the valley just reaches the breakup before sunset. However, this is only a crude approximation since

slope winds export energy out of the valley as well and, hence, reduce $Q_{\mathrm{prov}}$. The exported heat leads to an increase of $\Theta_E$, the potential temperature above the valley, in time for which reason $Q_{\mathrm{req}}$ increases as well. Hence, $B < 1$ is typically required for the breakup when calculating $B$ according to Eq. 9 (Leukauf et al., 2015). The breakup parameter can be used to describe the combined effect of stability and forcing on the total tracer mass export out of the valley. Figure 9b illustrates that this export, calculated for all simulations, is only a function of $B$. The corresponding exponential curve fitted to this data is given by

$$m_{\mathrm{tot}} = a \exp(-bB) + c, \tag{10}$$

with $a = 82.54$, $b = 2.098$ and $c = 6.79$. The export of tracer mass, which reaches values of 80 % for S3N06, decreases very fast with increasing $B$ and passes the 50 % mark at $B \approx 0.3$. At $B = 1.0$ it has decreased to about 15 %. Hence, an early breakup (i.e., $B << 1$) is required for an effective venting by convection.

## 4   Discussion

In general, an initially stably stratified valley atmosphere which is heated from the surface can be divided into three volumes: a mixed layer, a slope wind layer and a stable core. However, their relative portion on the total valley volume is a function of initial stability, amplitude of surface heating and time.

There will be a transition of the valley atmosphere trough several regimes during the course of the day (Leukauf et al., 2015). Due to the large spatial inhomogeneities in mountainous terrain and the existence of thermally driven winds, the CBL evolution

over mountains is complex (De Wekker and Kossmann, 2015). The various volumes and their development in time are a part of this complexity. Usually, the criterion used to define these volumes relies either on the location of the vertical heat flux minimum or on a threshold value for the vertical $\theta$-gradient. Furthermore, the volumes and borders in between are constantly changing. This makes the exact definition and determination of the mixed layer, slope wind layer and stable core difficult. Three possible definitions for the SWL depth, that have been presented in Sec. 2, generally lead to rather similar results (cf. Fig. 2). As

expected, the largest differences occur shortly before the breakup, when the stable core is almost removed and borders between mixed layer and SLW become diffuse. The impact of using a definition of the SWL depth based on the $\theta$-gradient-criterion instead of the heat-flux-criterion is discussed below.

Between these different valley volumes, there is transport of mass through up-slope winds, entrainment, horizontal intrusions and recirculation (cf. Fig. 1). Most of the tracer mass emitted on the valley floor is transported into the SWL during the course

of the day, but it strongly depends on stability and forcing amplitude whether the tracer is predominately exported at crest height or transported horizontally into the stable core and recirculates (cf. Fig. 5). In particular, it is the horizontal intrusion





above the boundary layer, which causes the horizontal transport into the stable core and enables a recirculation from there back to the mixed layer. For this reason, the magnitudes of the associated fluxes depend to some extent on the definition of boundary layer height and slope wind layer depth. Comparing results shown in Fig. 4 with fluxes calculated using SWL depths according

to the $\theta$-gradient-criterion shows that the relative difference between the two calculation methods are between 14 and 18 % for $f_{12}$. The vertical flux $f_{13}$ in the middle of the valley is up to 70 % larger in the first two hours of simulation since the boundary layer height is lower using the $\theta$-gradient-criterion. This difference decreases to about 15 % at noon. Flux $f_{23}$ is also larger in the early morning hours and reaches about 25 % smaller amplitudes in the afternoon. The largest difference between the two definitions is the fact that volumes $V_2$ and $V_3$ vanish up to two hours earlier for the $\theta$-gradient-criterion. Hence, a larger

fraction of the fluxes at crest height $f_{24}$ and $f_{34}$ is represented in flux $f_{14}$. Fluxes according to the $w$-criterion have not been calculated since the slope wind layer depth is similar to the one of the heat-flux-criterion. Only minor differences are to be expected. Overall, the sensitivity of the fluxes to these criteria is smaller than the sensitivity to stability and forcing. Also, the total export of tracer mass at crest height is independent of these definitions.

   The horizontal and vertical transport of tracer has been studied for a wide range of stabilities and forcing amplitudes, but

only for a very idealized framework. Soundings with constant stabilities throughout the atmosphere are used to initialize the model, which is not very realistic. Multiple elevated inversions are common (Emeis et al., 2007; Rotach et al., 2015) and may cause secondary HIs which would contribute to $f_{23}$ and lead to more tracer being intruded into the stable core at higher levels. The development of HIs and, hence, the strength of $f_{23}$ is also related to the terrain geometry. Wagner et al. (2015a) have found secondary circulations within the valley (i.e., HIs) for steep and narrow valleys, but none for shallow valley geometries.

The chosen quasi-two-dimensional valley geometry, that neither allows for the development of along-valley winds nor for a plain-to-mountain circulation, is a major simplification. Wagner et al. (2015a) have studied the impact of these circulations, e.g., on the tracer transport as a function of the valley geometry. In their simulations typical air parcel trajectories start at low levels over the plain, enter the valley as a result of up-valley winds, are subsequently captured by slope winds and lifted above crest height from where they are recirculated towards the plain. This circulation is stronger for deeper and narrower valleys and

effectively removes pollutants out of the valley. It is conceivable that this three-dimensional transport towards the plain away from the atmosphere above the valley reduces the reimport of pollutants into the valley by compensating subsidence compared to a two-dimensional case. A corresponding weaker downward transport of tracer into the valley in a 3D simulation compared to a 2D one has been noticed by Lehner and Gohm (2010).

   Due to the symmetry of the problem considered in this study, the two slope wind layers have been treated as a single volume.

In reality, asymmetric flow structures have been reported, e.g., in case of asymmetric solar forcing (Harnisch and Gohm, 2009; Gohm et al., 2009) and valley curvature (Weigel et al., 2007). Along-slope variations of the SWL due to changes in slope angle and soil properties are also neglected, although strong gradients in these properties could cause HIs as well (Lehner and Gohm, 2010). However, for the total export of tracer mass, the detailed structure within the valley is less important. Vertical fluxes associated with two asymmetric slope wind layers will probably export on average as much pollutants as those of two

symmetric layers. Further, a spatially constant forcing will cause on average a similar result as a complex, variable forcing as long as the bulk heat input into the valley atmosphere is comparable (Rotach et al., 2015). In other words, the exchange of



properties between the valley and the free atmosphere is mainly determined by bulk properties subsumed in the $B$ parameter rather than by spatial and temporal variability of stratification and surface heating. Hence, we believe that the highly idealized approach chosen in this study is suitable for drawing more general conclusions.

We have shown that the export of tracer out of the valley is well described by our breakup parameter $B = Q_{\mathrm{req}}/Q_{\mathrm{prov}}$ (Fig. 9b). Princevac and Fernando (2008) introduced a similar parameter, $B_{\mathrm{PF}} = \frac{N^3 H^2}{q_0}$, that essentially describes the ratio between the static stability of the valley atmosphere and the surface sensible heat flux[1]. Here we recall that the stability is proportional to $Q_{\mathrm{req}}$ and the surface sensible heat flux proportional to $Q_{\mathrm{prov}}$ (Eq. 8). Based on laboratory experiments, Princevac and Fernando (2008) concluded that the destruction of the cold-air pool in a valley is determined by convection for low values
of $B_{\mathrm{PF}}$ and by horizontal intrusions for high values of $B_{\mathrm{PF}}$. Similarly, we found that horizontal intrusions occur predominantly for large values of $B$. However, this is also the situation when a complete breakup of the valley inversion does not occur. Hence, it appears that horizontal intrusions alone are not efficient enough to neutralize the stratification in a valley.

It is clear that thermally driven winds provide an effective venting mechanism. The number of turnovers of valley air mass during daytime is between zero and five, depending on the forcing amplitude (Leukauf et al., 2015) and similar numbers have
been reported in the literature (e.g., Henne et al., 2004; Weigel et al., 2007; Wagner et al., 2015a). For a review of these studies see Rotach et al. (2015). We have shown that in addition to the forcing amplitude, stratification is another crucial parameters to determine the efficiency of vertical exchange processes, which is in good agreement to the work of Segal et al. (1988), Whiteman and McKee (1982) and Chemel et al. (2016). However, we found in this study that for a passive tracer, emitted at the valley floor at a constant rate, only up to about 80 % leaves the valley atmosphere. This is due to the redistribution of
tracer mass by HIs and the reimport at crest height. Consequently, the exchange efficiency as discussed so far (given turnover numbers of up to five) gives a too optimistic picture concerning the actual exchange of tracer mass. The breakup parameter $B$ is suitable to describe the mutual effects of stability *and* forcing for the vertical export of tracer mass on a daily basis (cf. Sec. 3e). For large values of $B$, most of the tracer mass stays not only inside the valley, but also mostly in the mixed layer $V_1$, which may occupy a relatively small fraction of the valley volume (cf. Fig. 6). Hence, tracer concentrations close to the valley
bottom are higher for larger values of $B$. Using an atmospheric sounding at sunrise, knowledge on the valley geometry and an estimation of the average surface sensible heat flux, one can compute the breakup parameter $B$ (Eq. 9) and consequently the fraction of the emitted pollution that is exported to the free atmosphere during daytime.

## 5    Conclusions

The evolution of a daytime valley boundary layer and the transport of a passive tracer between tree different valley sub-volumes
(mixed layer, slope wind layer, stable core) and the free atmosphere is investigated by means of large eddy simulations. The model is initialized with a constant Brunt-Väisälä frequency and the surface sensible heat flux is prescribed by a sine-shaped function. A large set of simulations have been performed for initial Brunt-Väisälä frequency $N$ ranging between $N = 0.006$ and $0.02 \ \mathrm{s}^{-1}$ and for surface heat flux amplitudes between 62.5 and 375 W m$^{-2}$. These are the main results:

---

[1]More specifically, $N$ is the Brunt-Väisälä frequency, $H$ is the inversion height and $q_0$ the buoyancy flux at the surface.





- A horizontal intrusion forms above the mixed layer at the transition zone between the slope wind layer and the stable core. This phenomenon is in agreement with the circulation described by Vergeiner's idealized slope-flow model (Vergeiner, 1982; Vergeiner and Dreiseitl, 1987) which predicts horizontal air-mass export from the slope wind layer as a result of vertically increasing ambient stability between the mixed layer and the stable core. By this circulation, tracer mass released at the valley floor is intruded into the stable core.

- For a given heat flux amplitude, the efficiency of the vertical tracer mass transport strongly depends on the static stability. For example, for a typical forcing amplitude of 125 W m$^{-2}$, the net export of tracer mass from the valley to the free atmosphere during the course of the day may range between 10 % and 65 % depending on whether the atmospheric stability is considerably higher or lower than the value of the standard atmosphere ($N = 0.01$ s$^{-1}$), respectively. About 40% is transported from $V_2$ to $V_3$ whereof half stays in the stable core and the other half is recirculated into the mixed layer.

- The vertical transport of tracer mass from the mixed layer into the slope wind layer primarily depends on the forcing amplitude, whereas the horizontal flux from the slope wind layer into the stable core and the export at crest height depend on both forcing amplitude and initial stability.

- There is a similarity between simulations with different stabilities and forcing amplitudes in the sense that a combination of weak forcing and weak stability leads roughly to the same export of tracer mass during the course of the day as a combination of strong forcing and strong stability.

- A so-called breakup parameter $B$ has been introduced that is able to describe the combined effect of stability and forcing on the total export of tracer mass. It is defined between the ratio of energy required to neutralize the stratification in the valley and the energy provided by the surface sensible heat flux over the course of the day.

- An early breakup of the valley inversion is required for an effective venting of pollutants. Half of the tracer mass released at the surface is exported for $B \approx 0.3$, whereas only about 15 % is exported for $B = 1$.

Although this study is limited by the choice of idealized assumptions, i.e., quasi-two dimensional topography, initially vertically constant stratification and horizontally homogeneous surface heating, it provides an overview of the magnitudes of pollutant transport for a wide variety of stabilities and forcing amplitudes. Also, we propose a single parameter, which describes the total export of tracer mass over the course of the day. In future studies, it would be desirable to test the dependence of pollutant venting on the breakup parameter for different valley topographies, including 3D geometries, and to clarify whether the export of other quantities like heat and moisture could be described by a similar relation.

*Author contributions.* D. Leukauf designed the numerical experiments, carried them out and prepared the manuscript. A. Gohm and M. W. Rotach provided suggestions for the design of the experiments, recommended relevant literature, discussed the results with the main author and contributed to the manuscript by critical comments on text and figures, and proof reading.

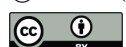



*Acknowledgements.* This work was supported by the Austrian Science Fund (FWF) under grant P23918-N21, by the Austrian Federal Ministry of Science, Research and Economy (BMWFW) as part of the UniInfrastrukturprogramm of the Research Focal Point Scientific Computing at the University of Innsbruck and by a PhD scholarship of the University of Innsbruck in the framework of the Nachwuchsförderung 2015. Computational results presented have been achieved in part using the Vienna Scientific Cluster (VSC).



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





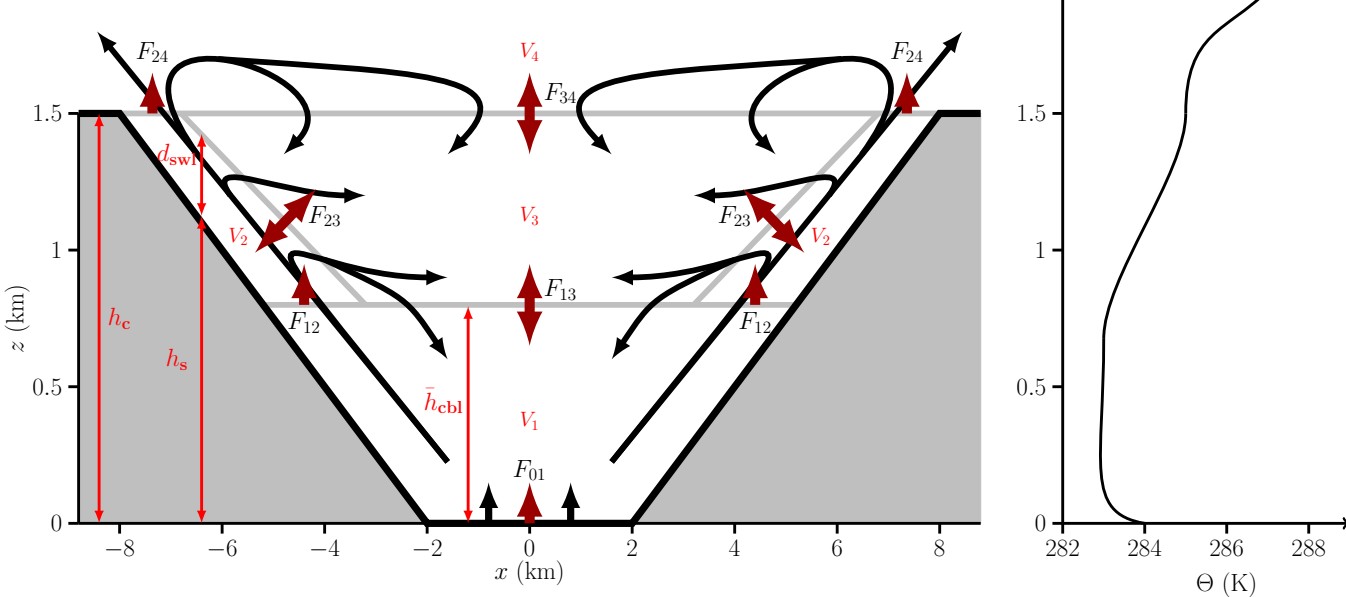

**Figure 1.** Valley topography with a schematic representation of volumes and tracer fluxes in between with and a typical profile of potential temperature at the center of valley after some hours of model integration. The valley is homogeneous in $y$-direction and is 10 km long. The valley volume is split into three time dependent parts (indicated by grey lines): the mixed layer ($V_1$), the slope wind layer ($V_2$), and the stable core ($V_3$). The volume of the free atmosphere ($V_4$) is assumed to be time-invariant. Notice that $V_2$ is the sum of the two slope wind layers on the two valley sides. Shown is regime 1 for well defined volumes. See text for explanation of regime 1 and 2. Black arrows illustrate some typical tracer trajectories. Thick red arrows represent bulk tracer mass fluxes between the volumes. Single headed arrows are used for bulk fluxes which are predominately directed upwards while double headed arrows represent integrals of local fluxes which change sign either in time or along the interface between volumes.





**Figure 2.** Vertical cross sections at 09 LT (left column) and 12 LT (right column) for the simulation S1N10. Shown fields in (a) and (b) are instantaneous cross-valley wind component (m s$^{-1}$) as colour contours and instantaneous potential temperature as black contours with 0.5 K increment at $y = 5$ km. The same variables, but averaged in time and along the $y$-axis, are shown in (c,d) together with wind vectors. The averaged and normalized tracer mass density is shown as $\log_{10}(\langle \bar{\rho}_{tr} \rangle / \max(\langle \bar{\rho}_{tr} \rangle))$ with colour contours in (e,f), along with potential temperature and wind vectors as in (c,d). The three different boundary layer and slope wind layer heights are shown as yellow lines. They are based on the heat-flux-criterion (solid line), the $\theta$-gradient-criterion (dashed line) and the $w$-criterion (dotted line). See text for further explanation.





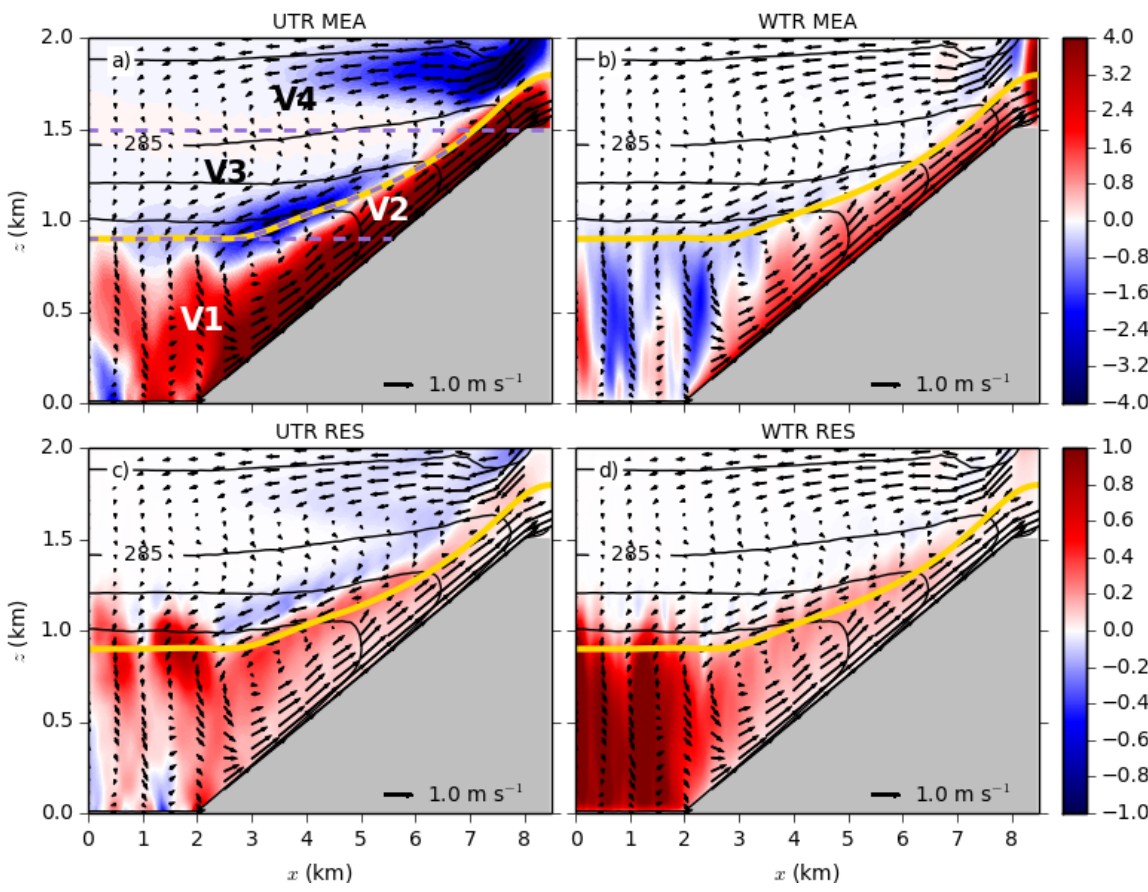

**Figure 3.** Horizontal and vertical tracer mass fluxes for simulation S1N10 at 12 LT. Shown are (a) cross-valley mean (UTR MEA), (b) vertical mean (WTR MEA), (c) cross-valley turbulent resolved (UTR RES) and (d) vertical turbulent resolved (WTR RES) tracer mass fluxes as colour contours. All fluxes are normalized by the tracer mass flux at the surface. The colorbars to the right apply to each row. Notice that the color scale in (c,d) is by a factor of four smaller than in (a,b). Potential temperature, wind vectors and the boundary/slope wind layer height according to the heat-flux-criterion are as in Fig. 2. The volumes V1 to V4 are also shown in (a). The purple dashed lines denote the boundaries between these volumes.





**Figure 4.** Normalized bulk tracer mass flux for a set of S1 simulations with five different static stabilities: (a) N06, (c) N08, (d) N10, (e) N12 and (f) N16. The tracer mass fluxes are integrated at the boundaries between different volumes and normalized by the spatially integrated tracer mass flux at the surface $F_{01}$. The sketch in (b) shows the position of different volumes and associated bulk fluxes. The left half represents a valley atmosphere before the breakup (regime 1) whereas the right half shows the situation after the breakup (regime 2). A grey dashed line indicates the change from regime 1 to 2. Colours of arrows in (b) are the same as the colours of the corresponding lines in the other panels, whereby solid (dashed) arrow shafts correspond to solid (dashed) lines.





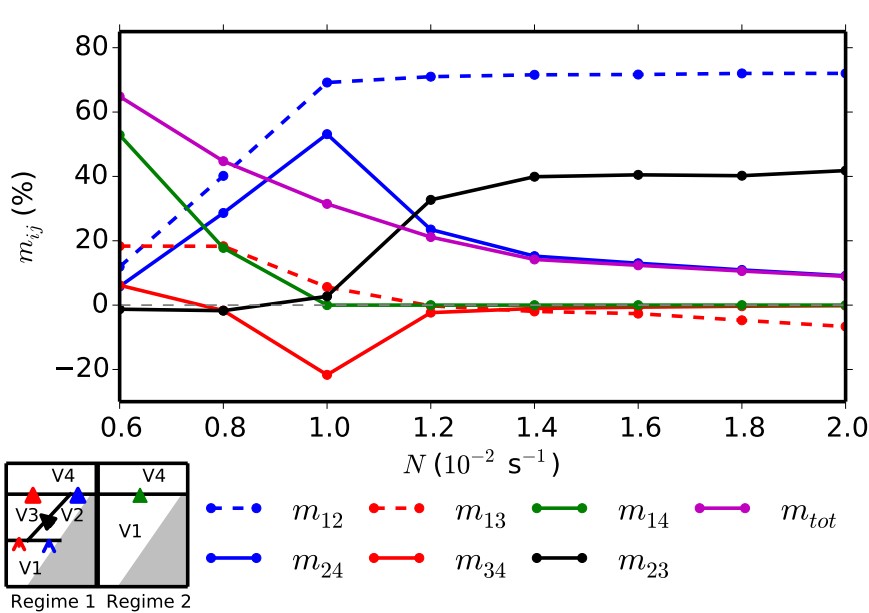

**Figure 5.** The total mass of tracer passing each interface between two volumes $V_i$ and $V_j$ and between 06 and 18 LT, normalized by the total mass of tracer released at the valley floor, i.e., $m_{ij} = M_{ij}/M_{01}$ as a function of stability for the S1 forcing. The total exported tracer mass over the course of the simulation is $m_{tot} = m_{24} + m_{34} + m_{14}$. In the bottom left is similar to Fig. 4b, except that the open arrowhead corresponds here to dashed lines.





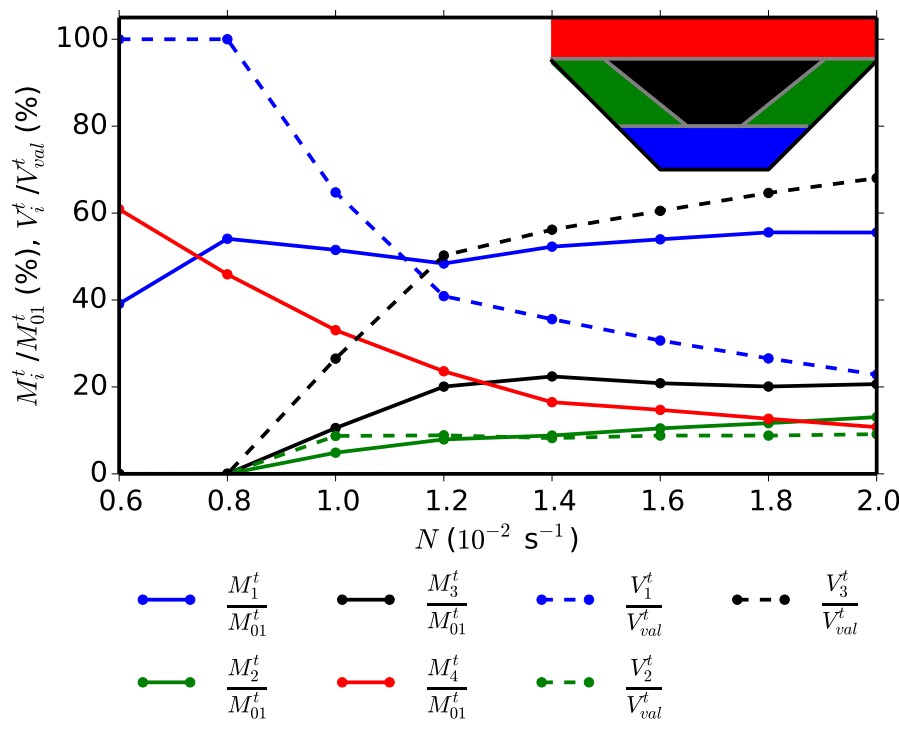

**Figure 6.** Fractions of tracer mass in four different volumes ($V_1$ to $V_4$) as solid lines and size of the volumes $V_1$, $V_2$ and $V_3$ relative to the valley volume $V_{val}$ as dashed lines one hour before the end of the simulation (17 LT) as a function of stability for the S1 forcing. Valley volumes are sketched in the top right corner in colors corresponding to the lines.





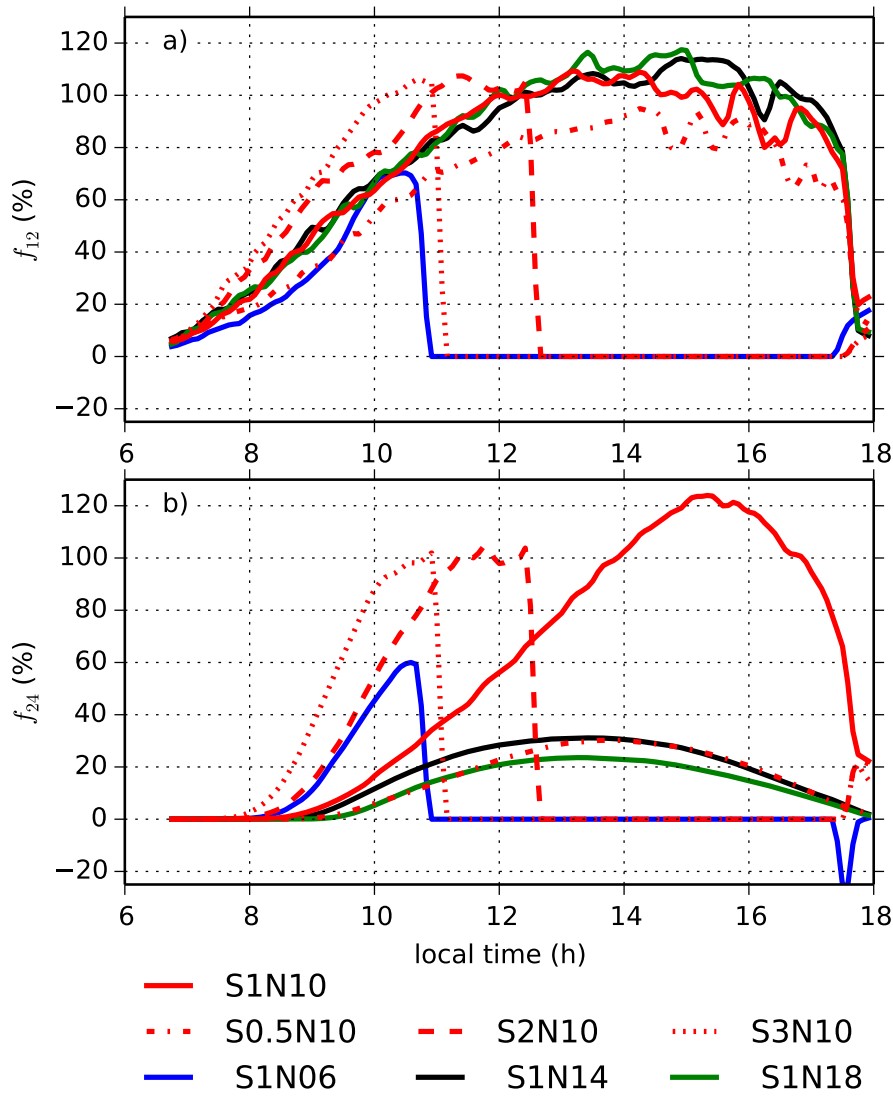

**Figure 7.** Tracer mass flux (a) $f_{12}$ and (b) $f_{24}$ for N10 and four different forcing amplitudes (S0.5N10, S1N10, S2N10, S3N10) as well as for S1 and four different stabilities (S1N6, S1N10, S1N14, S1N18).





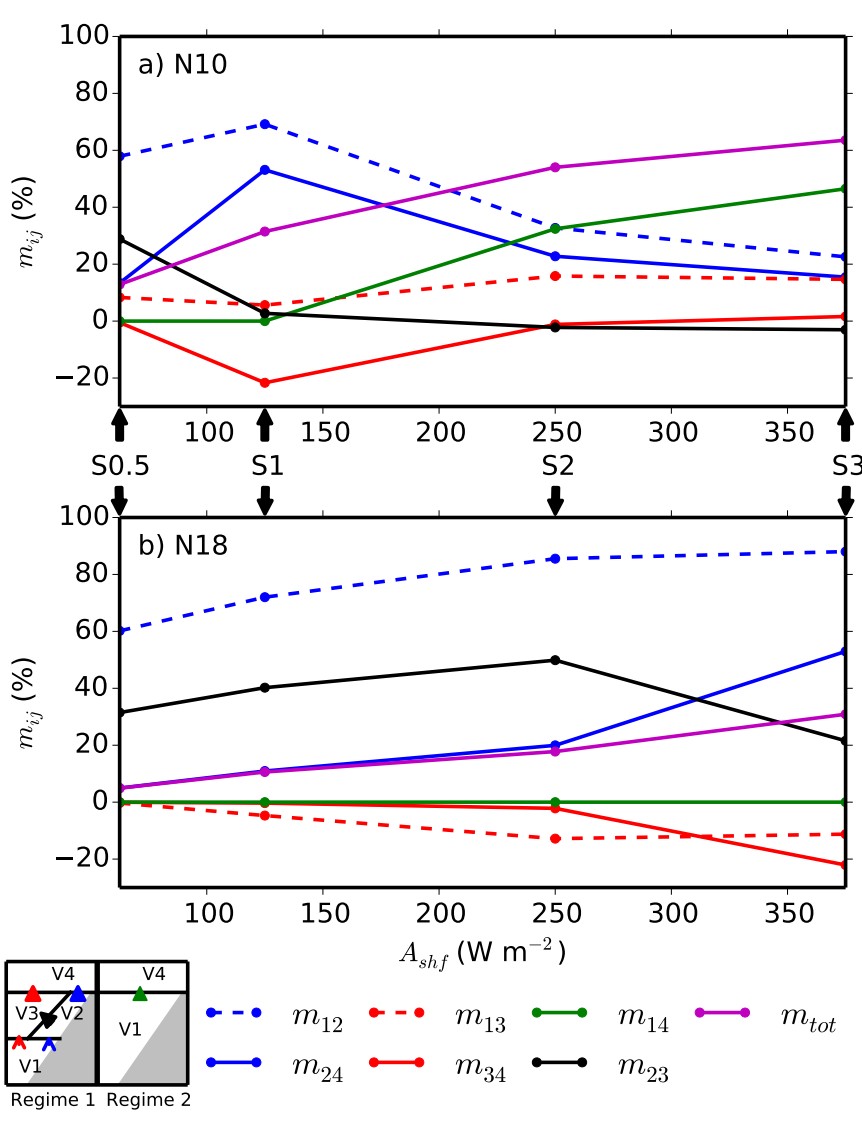

**Figure 8.** As in Fig. 5, but as a function of the forcing for (a) N10 and (b) N18.





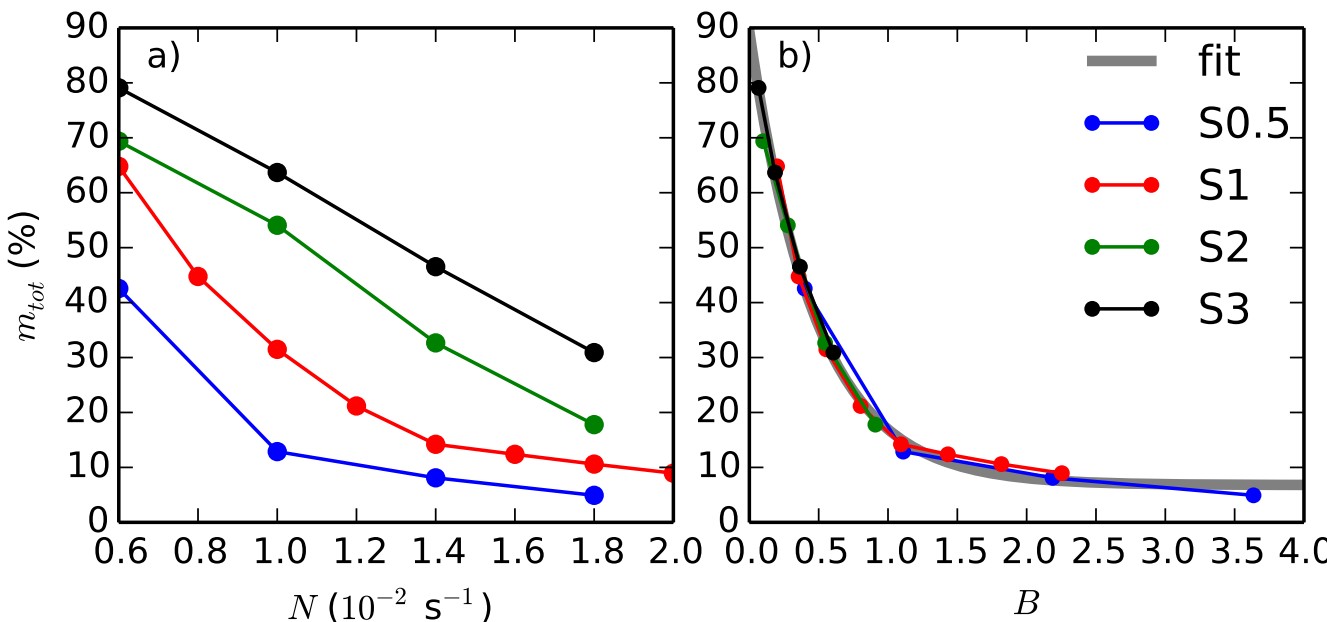

**Figure 9.** Total tracer mass exported out of the valley atmosphere at crest height normalized by the total tracer mass released at the surface as a function of (a) Brunt-Väisälä frequency and (b) breakup parameter for various different forcing amplitudes. A fit to the data according to Eq. 10 is shown in (b).