# Peer review of "Quantifying horizontal and vertical tracer mass fluxes in an idealized valley during daytime"

_Atmospheric Chemistry and Physics, 2016_

## Referee Comment (RC1) · Anonymous Referee #2 · 16 Jun 2016

**Review of 'Quantifying horizontal and vertical tracer mass fluxes of a daytime valley boundary layer' by Leukauf et al.**

The authors investigate the transport and mixing of pollution in an idealized valley during daytime using large-eddy simulations with WRF. The purpose of the manuscript is to quantify the horizontal and vertical tracer mass fluxes in dependence of surface sensible heat flux and initial stratification. Several sensitivity tests were performed systematically varying the surface sensible heat flux amplitude and the Brunt-Väisälä frequency. The valley volume is distributed in 3 different volumes and tracer mass fluxes between these volumes and the total exported tracer mass out of the valley volume are calculated. The total exported tracer mass significantly drops when increasing atmospheric stability and decreasing the surface sensible heat flux. The authors propose a single parameter to describe the combined effect of initial stratification and surface forcing on the total export of tracer mass.

The transport of pollution in valleys is an important research topic which has been studied extensively in the past, but still needs further investigations. The authors specifically address the role of horizontal fluxes from the slope wind layer into the stable core of the valley. Although the assumed atmospheric conditions and valley geometries are highly idealized and not necessarily realistic, I believe that the manuscript provides interesting results. However, I have several minor and some major comments regarding the definitions of valley volumes and the clarity of the result section. The presentation quality is mostly good too excellent, but there are multiple English spelling and grammar flaws and I recommend having the manuscript checked by a native speaker. In my opinion the manuscript is suitable for publication in ACP provided that the authors conduct a substantial review considering my comments given below.

**Major comments**

1. Definition of volumes: While I can follow the defintion of volume V1, I have concerns about the definitions of the other volumes. The authors define everything above the level of the valley ridge as free atmosphere (V4). However, slope winds and a CBL, which forms on the plateau at ridge height, occur above this level. They are surely not part of the free atmosphere but rather part of a boundary layer affected by the terrain. Thus, the conclusion that the total export of tracer mass into the free atmosphere is described by fluxes at the tops of V3 and V2 is not correct in my opinion. A possible solution could be to simply define V4 as the volume outside the valley atmosphere, whereat the valley atmosphere goes up to the valley ridge.

In Fig. 1 it seems like a neutrally stratified layer forms above the stable core (V3) topped by an inversion. Maybe a definition of the top of V3 taking into account the level of this neutrally stratified layer might be more appropriate than simply using a constant upper level for this volume.

2. Readability and clarity of the result section: I understand that the evolution of the tracer mass fluxes in the different simulations is very complex and requires a detailed analysis. The authors divide the analysis in 5 subsections and describe the different aspects in great detail in the respective sections. However, the amount of details significantly affects the readability and clarity of this section and makes it rather hard to follow for the reader. Especially section 3.2, 3.3 and 3.4 are difficult to read. Thus, I strongly encourage the authors to revise these sections and to focus on the most important features instead of describing everything in detail. This would improve the manuscript a lot.

**Minor comments**

Title: "Quantifying horizontal and vertical tracer mass fluxes in (?) a daytime valley boundary layer"

Abstract: The abstract is rather long (especially the part after l. 12) and the authors should consider shortening it by reducing the number of detailed information (specific surface heat flux or stability, percentages). Instead they should focus on the most important results.

l. 19-20: How can the authors quantify a transport to the stable core when a complete neutralization of the valley atmosphere is reached?

l. 22: "The total export of tracer mass out of the valley atmosphere …"

l. 29: How is the valley atmosphere exactly defined? Up to ridge height?

l.31: stably-stratified instead of stable as stable can also refer to stable in time

l. 32: The authors use buoyancy frequency as well as Brunt-Väisälä frequency for the same parameter.

l. 38: "but" does not make sense here.

l. 39-40: The aims of the study do not belong here. They rather belong to the end of the introduction where they are already mentioned in l. 91-93.

l. 51-52: How is turbulence generated when the HI penetrates the stable core? Due to shear? Please clarify.

l. 66 and l. 70: The abbreviation for convective boundary layer has to be given at its first occurrence and should be used consistently thereafter. The same applies to other abbreviations (e.g. HI).

l. 70: Per definition mixed layer and CBL are not the same, as the CBL consists of a surface layer, mixed layer and entrainment layer. If the authors choose to use both terms CBL and mixed layer (ML)

I recommend to distinguish clearly between both terms. However, I do not see the point of using both terms as the authors mostly refer to CBL in their manuscript.

l. 73-76: Other important mechanisms for pollution transport are the upward transport of polluted air over the mountain ridges due to slope winds (mountain venting, e.g. Fast and Zhong 1998) and the horizontal downstream advection of polluted air from over the ridges with the large-scale flow (advective venting, e.g. Kossmann et al. 1999). These mechanisms substantially affect the evolution of the boundary layer over mountainous regions and in particular over valleys and should be mentioned.

l. 77-84: This detailed paragraph on plain-to-mountain winds is not relevant for the study.

l. 89-90: The order of the references is not uniform.

l. 93: It would be interesting to get some information on the chosen setup already in the introduction. For example, what are the valley dimensions and what was the motivation for that? Are they similar to a typical alpine valley? What is the range of surface forcing? Are the conditions typical for summer, winter, fall conditions?

l. 93: The authors also investigate the total export of pollutants out of the valley atmosphere which would be interesting to mention here.

l. 93-95: This sentence does not fit here. It rather belongs to the motivation at the beginning of the introduction.

l. 97: a detailed list of the presented results would be helpful for the reader.

l. 109: Motivation for valley dimension? See previous comment

l. 117-118: Motivation for amplitude of surface sensible heat flux?

l. 123ff: How realistic is the chosen setup? The authors chose a uniformly stratified atmosphere at rest for initialization. I do not see how this resembles any realistic conditions. What about a valley floor inversion? I suppose this would significantly influence the results. The authors should at least discuss this.

l. 130: A table summarizing the different runs would help for clarity.

l. 142: What is the reason for 41 minutes averages?

l. 147: Please specify the difference between volumes and bulk fluxes.

l. 148: As long as the valley atmosphere is …

l. 154: What does more useful mean?

l. 157: What time is sunrise? At initialization time? Please clarify.

l. 159-160: How is the breakup detected? I believe that once the stable core is eroded the heat flux criterion detects the inversion around ridge height as h_cbl.

l. 166: Why is that useful?

l. 174-175: Are the budgets for the different volumes closed? What about sub-grid scale fluxes? Did the authors quantify these?

l. 196: Do the authors have an explanation for the large difference?

l. 216: What are the similar characteristics?

l. 232: It is not so much a failure but rather results from the criterion itself.

l. 234: The heat-flux criterion is not used for the upper boundaries of V2 and V3. The authors should be more precise here.

l. 235: How is the boundary between V1 and V2 determined?

l. 243: What does in the morning mean? I guess heating starts once the simulations are started? CBL instead of mixed layer

l. 244: the description of the SWL is not necessary as it is already introduced. Why is the SWL and CBL not indicated in Figs. 2a, b?

l. 247: here and thereafter: once the abbreviation HI is defined it should also be used.

l.252: description of HI not necessary anymore.

l. 254: CBL instead of mixed layer.

l. 260-263: Did the authors investigate the variability of HI for the different output times? This might be interesting to know.

l. 264-265: How did the authors detect the turbulent mixing?

l. 271-274: This rather belongs to the description of the other simulations in the next section.

l. 267-268: Where does this second and stronger circulation come from? Is this symmetric, i.e. does it occur on the other ridge as well? This circulation is right at the boundary of the model domain. Is this a numerical boundary effect? Why did the authors choose the boundaries of the model domain so close to the valley? Did the authors perform sensitivity runs with other domain boundaries? According to the definition of the authors this circulation occurs in the free atmosphere, which is somewhat contradictory as it obviously is a boundary layer process (see major comment 1).

l. 278: into the CBL

l. 284: If regime 2 is never reached in S1N10 it can't be reached earlier or later in other runs.

l. 290: bulk or volume fluxes? See previous comment.

l. 293: the authors use SWL and V2 synchronous (here and at several locations in the text). It would improve readability if the authors used only one term.

l. 312: How can the normalized flux exceed 100%? The definition of the top of volume V1 during regime 2 is not clear to me. How deep was the neutrally stratified layer? The authors stated that is exceed the ridge height (l.272). See also major comment 1.

l. 338-339: This sentence is irrelevant.

l. 341: How is the total tracer mass calculated? Between 6 and 12 LT or between 6 and 18 LT as it is stated in Fig. 5?

l. 364: Why does the surface sensible heat flux decrease sharply at 18 LT? I thought it follows a sine curve?

l. 374: Dependence on forcing amplitude and initial stability?

l. 378-379: english: For simulations with different than ???

l. 381: As far as I understand $F_{12}$ is defined at height $h_{cbl}$ and the horizontal extent of the area reaches from the bottom of the slope ($x = 2$ km) to the slope. With different surface heating I expect the height $h_{cbl}$ and thus the horizontal area to vary strongly. How does this affect the fluxes and what does this mean for the comparability of the different runs? Do the authors consider this in the analysis?

l.390: What is "usual meaning"? I do not understand lower end of the SWL? Please clarify.

l. 393: SWL into the free atmosphere. See major comment 1.

l. 423: Change the title of this subsection so that it reflects its content better. E.g. Definition of a dimensionless parameter

l. 439: English: not further conclusions…

l. 441: … important for what?

l. 446-447: .. with strong stability and/or weak forcing … implies weak stability and/or strong forcing …

l. 448: As B is defined as the ratio between required and provided energy it does not depend directly depend on the breakup time. It is rather a measure of total mixing. Only its sign gives some information if the breakup is reached (B<1) or not. Thus, I do not understand why the authors call it breakup parameter. In my opinion, this is misleading. Something like "mixing parameter" or "export parameter" would be more appropriate.

l. 459ff: The discussion section should be improved. At the moment it is rather long and repeats a lot of previous studies (e.g. l. 495-503 or l. 523-526) but often misses a critical discussion of the findings of this manuscript to other studies (e.g. l. 489-494).

l. 464: What is inhomogeneous? The terrain or the atmosphere?

l. 463-488: The sensitivity of the tracer fluxes to the different definitions of the volumes is very interesting. The discussion would benefit from an additional figure visualizing the sensitivity.

l. 495-503: this is pretty much a summary of previous studies. It would be more interesting to discuss the aspects related to terrain geometry (depth, width, slope angles, …) which have not been considered in this study and maybe speculate about their impact.

l. 553: referring to V2 and V3 in the summary is not ideal, as a reader only reading the summary would not understand it.

l. 555-558: Some more details on the how?

l. 570-571: Testing the dependence on realistic atmospheric stratifications (i.e. several elevated inversions) would be interesting too.

Fig. 2: How long are the time averages? Are the wind vectors time averaged as well?

Fig. 4: Are the shown fluxes averaged in time or instantaneous? It would be helpful to include the total export into V4 ($f\_24+f\_34$).

Fig. 8: "…. Of the forcing amplitude, $A\_shf$, …"

**References:**

Fast JD, Zhong S (1998) Meteorological factors associated with inhomogeneous ozone concentrations within the Mexico City basin. J Geophys Res 103(D15):18927–18946

Kossmann M, Corsmeier U, De Wekker SFJ, Fiedler F, Vögtlin R, Kalthoff N, Güsten H, Neininger B (1999): Observations of handover processes between the atmospheric boundary layer and the free troposphere over mountainous terrain. Contr Atmos Phys 72:329–350

---

## Referee Comment (RC2) · Anonymous Referee #1 · 7 Jul 2016

This paper describes the results of a series of idealized large eddy simulations to quantify the vertical and horizontal transport of tracers within various layers of the daytime valley atmosphere. While the results are not very surprising, the paper is interesting and overall well-written but some clarification is needed in certain locations. Some comments are provided below.

1) When reading the paper, I had the impression that there was quite some overlap with previous papers published by the Innsbruck group. Some clear(er) explanations on how the current study follows up on these previous papers and in what respects it is similar, would be desirable.

2) The introduction states the general broad aim of this paper but a strategy on how to address the goals of the paper appears to be missing in the Introduction.

[Figure]

3) line 67: remove 'a' in 'a complex interactions"

4) line 90: "...the role...have..." should be ...the role...has..."

5) line 90: in terms of what has the role of intrusions been evaluated before?

6) line 111: Misleading sentence. The valley is not really 17 km broad, the model domain is. Rephrase.

7) line 119: runs for 12 hours, not run.

8) line 124: Define A_shf

9) line 123-130: for clarity/reference, it would be good to include a table with the listing of the various simulations

10) line 131: is the passive tracer released at every gridpoint?

11) line 142: "over 41 minutes". Seems like a weird number. Why 41 and not e.g. 30 or 60 minutes?

12) line 156: fourth volume, not forth

13) line 166: "It is useful to....". Explain why it is useful.

14) line 185 and following: I had to read the explanation how F23 is calculated several times and it is still not very clear. Any way the authors can make the explanation clearer would be appreciated.

15) section 2.4: it is very good the authors pay attention to their definition of CBL heights over valley and slope. I am not convinced though that the slope wind layer is synonymous as the slope CBL as the authors suggest. Better would be to just call it slope CBL rather than slope wind layer.

16) section 3.1 seems rather qualitative. Include some more quantitative information including e.g. the speeds and height of the slope flows.

17) line 378: "For simulations with different than..." sounds awkward. Rephrase.

18) line 450: the authors mention the export heat here. This made me think whether the authors could address, perhaps in the discussion section, if they expect heat and moisture to be transported in a similar way as the passive tracers. The evolution of temperature and humidity profiles in the valley could address this issue.

19) I like the discussion of some issues/limittaions with the idealized simulations in the discussion section. I can think of many more than the authors discuss but I realize that discussing/mention these may weaken the study. Some mentioning of the limitations in the abstract would also be appropriate.

20) line 562: It is defined "as", not "between"

21) line 568: the single parameter does not describe the total export of tracer mass, it rather is indicative of the total export.

22) Fig. 3: Include the unit of the numbers in color bar

---

## Author Comment (AC1) · 26 Aug 2016

**Response to the Reviewers' Comments (acp-2016-350)**

Daniel Leukauf, Alexander Gohm and Mathias W. Rotach

August 26, 2016

We thank all reviewers for their clear and helpful comments. All sections have been revised. Special attention has been given to the sections 'Introduction', 'Methods' and 'Discussion' since most questions were directed to these sections. A paragraph discussing if and how the results of this study can be transferred to fluxes of other quantities like moisture has been added to the discussion. We also clarified formulations regarding the methodology and improved the introduction to better explain how this study is related to previous work done by our group. Furthermore, as anonymous reviewer #2 suggested, we asked a native English speaker to read the manuscript in order to detect all language related flaws.

In this document, comments of the reviewers are written in italic style and our replies in normal font style. A copy of the revised manuscript in which our changes are coloured in red is provided as well. Only major changes have been marked. Minor corrections (typos, minor reformulations, changes like 'slope wind layer' to 'SWL') are not marked.

In the following we give a point-by-point assessment of all the reviewer comments.

**Review 1**

**General Comments:**

*1) When reading the paper, I had the impression that there was quite some overlap with previous papers published by the Innsbruck group. Some clear(er) explanations on how the current study follows up on these previous papers and in what respects it is similar, would be desirable.*

We agree that it is important to highlight overlaps and links to previous studies. Studies from the Innsbruck group which are relevant for this study have been cited and discussed, but we see that a more explicit explanation of the connections is required.

We have modified the second-to-last paragraph of the introduction and have added:
The chosen quasi-two-dimensional valley geometry is a simplified version of the one used by Schmidli (2013). Initial atmospheric stability range from about half to twice the stability of the standard atmosphere and the forcing amplitude of 125 W m$^{-2}$ corresponds

approximately to the average amplitude of fluxes as observed during a 'radiation day' in the Riviera Valley in late August (Rotach et al., 2007). This reference forcing amplitude has been multiplied by the factors 0.5, 2 and 3. Studying both the impact of the initial stratification and forcing amplitude complements the studies of Wagner et al. (2014, 2015a,b) that are concerned with the impact of the terrain geometry. Including tracer mass transport naturally extends the work of Leukauf et al. (2015) which focuses on time of the inversion breakup. The inclusion of different initial stability conditions also extends the parameter range. By separating the valley atmosphere in sub-volumes representing the CBL, the slope wind layer and the stable core, we can calculate tracer mass fluxes between these volumes for a large number of stability/forcing combinations.

*2) The introduction states the general broad aim of this paper but a strategy on how to address the goals of the paper appears to be missing in the Introduction.*

We agree that an outline of the strategy is missing. It has been added to the introduction now (line 92ff). Please also refer to the response to comment 1.

**Specific Comments:**

*3) line 67: remove 'a' in 'a complex interactions'*
Done

*4) line 90: "...the role...have..." should be ...the role...has..."*
Done

*5) line 90: in terms of what has the role of intrusions been evaluated before?*
The sentence in question is indeed misleading. What we meant is that horizontal intrusions have been studied but the associated fluxes have never been quantified so far. We rephrase the sentence

'However, the role of intrusions in terms of mass transport have never been evaluated before.'

to:
'However, tracer mass fluxes associated with horizontal intrusions have never been quantified so far.'

*6) line 111: Misleading sentence. The valley is not really 17 km broad, the model domain is. Rephrase.*

The reviewer is correct. It is rather the domain itself which is 17 km broad. The sentence in question has been changed to:
In the horizontal, the domain consists of $340 \times 200$ grid points. With a mesh size of 50 m, this leads to a 17-km broad and 10-km long domain.

**7) line 119: runs for 12 hours, not run.**
Done

**8) line 124: Define $A_{shf}$**
The variable $A_{shf}$ is now introduced a few lines above:
The surface sensible heat flux is prescribed using a sine-function with amplitudes $A_{\mathrm{shf}} = 62.5$, 125, 250 and 375 W m$^{-2}$ and a period of 24 hours.

**9) line 123-130: for clarity/reference, it would be good to include a table with the listing of the various simulations**
A table providing an overview has been added (Tab. 1.).

**10) line 131: is the passive tracer released at every gridpoint?**
Only at lowest model level, over the valley bottom, but over the whole length of the valley. We have added 'at every grid point':
The passive tracer, that is used to measure the horizontal and vertical exchange rates, is released at the lowest model level at every grid point between $x =$ -2 km and +2 km, i.e., at the valley bottom and along the whole length of the valley.

**11) line 142: "over 41 minutes". Seems like a weird number. Why 41 and not e.g. 30 or 60 minutes?**
We have chosen this period following Schmidli (2013) and Wagner et al. (2014) and have used it also in Leukauf et al. (2015). The average in time is based on data which is sampled every full minute and includes data from the current time step plus/minus 20 minutes. We have added the following to the paragraph in question:
This corresponds to the minute of the timestamp plus/minus 20 minutes; this averaging interval was used in previous studies (e.g. Wagner et al., 2014), where it was demonstrated that at least 30 minutes are required to obtain reliable averages.

**12) line 156: fourth volume, not forth**
Typo corrected.

**13) line 166: "It is useful to....". Explain why it is useful.**
We have added an explanation and the passage now reads:
It is useful to normalize fluxes between volumes with the flux at the surface, i.e., $f_{ij} = F_{ij}/F_{01}$. By quantifying tracer mass fluxes between volumes as relative fluxes $f_{ij}$, the results are independent of the strength of the surface emission rate $F_{01}$.

*14) line 185 and following: I had to read the explanation how F23 is calculated several times and it is still not very clear. Any way the authors can make the explanation clearer would be appreciated.*

The explanation has been revised and should be clearer now.

*15) section 2.4: it is very good the authors pay attention to their definition of CBL heights over valley and slope. I am not convinced though that the slope wind layer is synonymous as the slope CBL as the authors suggest. Better would be to just call it slope CBL rather than slope wind layer.*

We agree with the reviewer that the slope wind layer and slope CBL are in principle different layers, but as we show in Fig. 2, they are almost identical between $h_{CBL}$ and crest height. For the sake of simplicity, we decided keep the denotation 'slope wind layer' but briefly discuss whether this is appropriate in the definition.

*16) section 3.1 seems rather qualitative. Include some more quantitative information including e.g. the speeds and height of the slope flows.*

We have added the slope wind layer heights and average along-slope wind speeds for 09 and 12 LT at x = 4 km and x = 6 km.

*17) line 378: "For simulations with different than..." sounds awkward. Rephrase.*

The sentence has been rephrased and reads now:

The overall structure of the valley atmosphere and the magnitudes of tracer fluxes between various volumes of the reference simulation are very similar to those from runs with a weaker or stronger forcing.

*18) line 450: the authors mention the export heat here. This made me think whether the authors could address, perhaps in the discussion section, if they expect heat and moisture to be transported in a similar way as the passive tracers. The evolution of temperature and humidity profiles in the valley could address this issue.*

We expect the transport of heat to follow by and large very similar patterns as those of the passive tracer. Especially the total export at crest height is very similar, but some very interesting differences have been identified. We have dedicated a whole paper to these issues which will be submitted very soon. The transport of moisture is, however, a more difficult process since condensation, cloud/radiation interaction and possibly rain and/or snow are expected to cause various complications. However, as long as there is no condensation, the transport of moisture is likely very similar to the transport of a passive tracer.

We address this in more detail in the discussion section now (line 541ff), but a detailed analysis would go beyond the scope of this paper.

**19) I like the discussion of some issues/limitations with the idealized simulations in the discussion section. I can think of many more than the authors discuss but I realize that discussing/mention these may weaken the study. Some mentioning of the limitations in the abstract would also be appropriate.**

Limitations of this study are now mentioned in the abstract.

**20) line 562: It is defined "as", not "between"**

Corrected

**21) line 568: the single parameter does not describe the total export of tracer mass, it rather is indicative of the total export.**

We have rephrased the sentence in question to:

A so-called breakup parameter $B$ has been introduced that combines the effect of stability and forcing on the total export of tracer mass.

**22) Fig. 3: Include the unit of the numbers in color bar**

The tracer mass fluxes shown in Fig. 3 have been normalized by the tracer mass flux at the surface as stated in the caption. It is therefore dimensionless.

**Review 2**

**General Comments:**

*1) Definition of volumes: While I can follow the definition of volume V1, I have concerns about the definitions of the other volumes. The authors define everything above the level of the valley ridge as free atmosphere (V4). However, slope winds and a CBL, which forms on the plateau at ridge height, occur above this level. They are surely not part of the free atmosphere but rather part of a boundary layer affected by the terrain. Thus, the conclusion that the total export of tracer mass into the free atmosphere is described by fluxes at the tops of V3 and V2 is not correct in my opinion. A possible solution could be to simply define V4 as the volume outside the valley atmosphere, whereat the valley atmosphere goes up to the valley ridge. In Fig. 1 it seems like a neutrally stratified layer forms above the stable core (V3) topped by an inversion. Maybe a definition of the top of V3 taking into account the level of this neutrally stratified layer might be more appropriate than simply using a constant upper level for this volume.*

We agree that the term 'free atmosphere' does not represent volume V4. It has been replaced by the term 'atmosphere above the valley'. It is true that a boundary layer evolves above the valley as well and $V_4$ could be separated into two volumes (a CBL above the valley and the free atmosphere above). However, a fixed reference height allows to compare the export of heat for different stability/forcing combinations while the fluxes at the CBL top above the valley are by definition very small. Presumably, only the height of this upper CBL would be different.

*2. Readability and clarity of the result section: I understand that the evolution of the tracer mass fluxes in the different simulations is very complex and requires a detailed analysis. The authors divide the analysis in 5 subsections and describe the different aspects in great detail in the respective sections. However, the amount of details significantly affects the readability and clarity of this section and makes it rather hard to follow for the reader. Especially section 3.2, 3.3 and 3.4 are difficult to read. Thus, I strongly encourage the authors to revise these sections and to focus on the most important features instead of describing everything in detail. This would improve the manuscript a lot.*

We thank the reviewer for this comment. We see now that too many details have found their way into this section and some of them distract from key features. We have revised section 3.2, removed less important details and streamlined formulations. In section 3.2 we have removed Fig. 6 and the paragraph describing it. Is was a rather complicated graph which required a lot of explanation and had too little relevance for the key findings of this paper. Only minor changes have been made in Sec.3.4 since the details given in this section are relevant for the key conclusions.

**Specific Comments:**

*3) Title: "Quantifying horizontal and vertical tracer mass fluxes in (?) a daytime valley boundary layer"*

We have changed the title to 'Quantifying horizontal and vertical tracer mass fluxes in an idealized valley during daytime'.

*4) Abstract: The abstract is rather long (especially the part after l. 12) and the authors should consider shortening it by reducing the number of detailed information (specific surface heat flux or stability, percentages). Instead they should focus on the most important results.*

Some of the less important information has been removed in favour of more generally valid statements. The abstract has been shortened as well.

*5) l. 19-20: How can the authors quantify a transport to the stable core when a complete neutralization of the valley atmosphere is reached?*

We agree that the sentence in question is misleading. After the breakup, it is of course no longer possible to quantify the flux into the stable core because it doesn't exist anymore. Fluxes into and out of the SC are therefore only evaluated as long as the stable core exists.

However, the abstracts was modified in response to comment 4 and the sentences in question have been removed.

*6) l. 22: "The total export of tracer mass out of the valley atmosphere ..."*

We included the words 'out of the valley atmosphere' as suggested.

*7) l. 29: How is the valley atmosphere exactly defined? Up to ridge height?*

There seems to be no consensus on how the valley atmosphere should be defined exactly. Schmidli and Rotunno (2010) have chosen ridge height + 100 m, Weigel et al. (2007) the height where the daytime along valley flow changes to the synoptic flow above the valley, which approximately coincides with the crest height. Other authors (i.e., Wagner et al., 2015a; Lang et al., 2015) define different boundary layer heights to separate the valley atmosphere from the atmosphere above the valley. In their case, the valley BL extends considerably out of the valley. We have chosen the ridge height as the upper boundary for the valley atmosphere and use therefore a geometric rather than a thermodynamic definition. The vertical trace-mass fluxes out of the valley are therefore comparable for different forcing/stability combinations (see our answer to comment 1).

*8) l.31: stably-stratified instead of stable as stable can also refer to stable*

*in time*

We agree and improved the wording.

**9) l. 32: The authors use buoyancy frequency as well as Brunt-Väisälä frequency for the same parameter.**

Is is indeed better to use only one term. We have decided to use 'buoyancy frequency' consistently.

**10) l. 38: "but" does not make sense here.**

"but" removed.

**11) l. 39-40: The aims of the study do not belong here. They rather belong to the end of the introduction where they are already mentioned in l. 91-93.**

We disagree. The first paragraph is the motivation for this study and we think that a very brief description of the goals does belong here. We have slightly rephrased the last sentence though.

**12) l. 51-52: How is turbulence generated when the HI penetrates the stable core? Due to shear? Please clarify.**

Indeed, shear is (by far) the primary cause of turbulence associated with the HI. There is also advection of turbulence from the slope wind layer. This is a much smaller, but relevant contribution.

We add this information and the sentence in question now reads:

However, turbulence is predominately generated through shear as the HI penetrates into the stable core. Additionally, turbulence is advected from the slope wind layer towards the center of the valley.

**13) l. 66 and l. 70: The abbreviation for convective boundary layer has to be given at its first occurrence and should be used consistently thereafter. The same applies to other abbreviations (e.g. HI).**

We use now the various abbreviations (CBL, HI and SWL) consistently. However, at the beginning of the section 'Discussion' and in the whole section 'Conclusion' they are written out again, since these chapters should be understandable without in depth reading of the introduction.

**14) l. 70: Per definition mixed layer and CBL are not the same, as the CBL consists of a surface layer, mixed layer and entrainment layer. If the authors choose to use both terms CBL and mixed layer (ML). I recommend to distinguish clearly between both terms. However, I do not see the point of**

*using both terms as the authors mostly refer to CBL in their manuscript.*

We agree and decided to refer consistently to the CBL.

**15) l. 73-76: Other important mechanisms for pollution transport are the upward transport of polluted air over the mountain ridges due to slope winds (mountain venting, e.g. Fast and Zhong 1998) and the horizontal downstream advection of polluted air from over the ridges with the large-scale flow (advective venting, e.g. Kossmann et al. 1999). These mechanisms substantially affect the evolution of the boundary layer over mountainous regions and in particular over valleys and should be mentioned.**

Both advective and mountain venting are now mentioned and briefly discussed (line. 79 ff). This new paragraph has replaced the one discussing the plain-to-mountain winds (see question 16).

**16) l. 77-84: This detailed paragraph on plain-to-mountain winds is not relevant for the study.**

We have rewritten the paragraph in question and reduced the details given.

**17) l. 89-90: The order of the references is not uniform.**

References put in chronologic order.

**18) l. 93: It would be interesting to get some information on the chosen setup already in the introduction. For example, what are the valley dimensions and what was the motivation for that? Are they similar to a typical alpine valley? What is the range of surface forcing? Are the conditions typical for summer, winter, fall conditions?**

We are giving now some information about the chosen valley dimensions and forcing, but we prefer to give details in the methods section (l 93 ff).

**19) l. 93: The authors also investigate the total export of pollutants out of the valley atmosphere which would be interesting to mention here.**

A good point. This fact is mentioned now.

**20) l. 93-95: This sentence does not fit here. It rather belongs to the motivation at the beginning of the introduction.**

The sentence has been moved to the first paragraph.

**21) l. 97: a detailed list of the presented results would be helpful for the reader.**

A detailed list is now provided (line 96ff).

**22) l. 109: Motivation for valley dimension? See previous comment**

The depth of the valley (1500 m) is comparable with the Inn Valley. These two valleys are, however, narrower than the valley we are using, which is more comparable with the northern Rhine Valley. The valley topography we have chosen is based on the REF2D topography used in Schmidli (2013). We have replaced the sine shaped slopes by linear slopes with the same maximum slope angle. Schmidlis REF2D topography has been used by multiple authors as well, which allows direct comparisons. This is now explained in the method section (line.119 ff).

**23) l. 117-118: Motivation for amplitude of surface sensible heat flux?**

The amplitude of the reference simulation (S1N10) has been set to a value which is just below the threshold to facilitate a complete breakup of the inversion. This way, variations of $N$ and $A_{shf}$ lead to situations in which the breakup happens either (much) earlier than 18:00 LT, or the heating is far too weak to erode the inversion. In other words, S1 in combination with N10 (standard atmosphere) is a good reference and starting point to explore the parameter space.

The range of the forcing has been inspired by (Rotach and Zardi, 2007), who have compiled average daily cycles of the surface sensible heat flux using 15 'sunny convective days with weak synoptic forcing' during summer for 7 different sites in the Riviera Valley. They report amplitudes of the surface sensible heat flux between 90 and 300 W m$^{-2}$ depending on the location.

The stability on the other hand is varied between a very low stability N6 (about half the stability of the standard atmosphere) and N20 twice the reference. Although much stronger stabilities are common in cold pools, this is not representative for a whole valley volume. Additionally, runs with strong stabilities (N16-N20) behave rather similar, hence extending the stability range towards even stronger stabilities wouldn't provide more insight.

We have added the following line to the manuscript:
These amplitudes cover the range reported for different sites at an Alpine valley during weak synoptic conditions (Rotach and Zardi, 2007).

**24) l. 123ff: How realistic is the chosen setup? The authors chose a uniformly stratified atmosphere at rest for initialization. I do not see how this resembles any realistic conditions. What about a valley floor inversion? I suppose this would significantly influence the results. The authors should at least discuss this.**

It is true that the initial conditions do not resemble very realistic conditions, but the aim of this study is to gain general insight, not to reproduce one particular weather situation. An atmosphere at rest fits fairly well during weak-gradient conditions, but the uniform stratification is a major, but widely used simplification (i.e., Schmidli and Rotunno, 2010; Catalano and Moeng, 2010; Serafin and Zardi, 2010; Wagner et al., 2014). Also, in Leukauf et al. (2015) we allowed a valley atmosphere to develop for 24 hours which led to a cold

pool topped by a residual layer. While there are indeed differences with regards to the evolution of the valley atmosphere, the time of the breakup differs independently of the forcing amplitude 'only' by about 1 hour. Also, using a more realistic sounding would question the generality of this study. We have expanded the paragraph discussing this issue in Sec. 4.

**25) l. 130: A table summarizing the different runs would help for clarity.**
Reviewer 1 had a similar request and a table summarizing the simulations has been added (Tab. 1).

**26) l. 142: What is the reason for 41 minutes averages?**
Please refer to the answer of question 11 from reviewer 1.

**27) l. 147:Please specify the difference between volumes and bulk fluxes.**
The title of the subsection seems to be a bit misleading. We discriminate between three types of tracer mass fluxes:

1) tracer mass fluxes are calculated at every grid point according to the method described in Sec. 2.2. Usually, we do not refer to this data in the article, as we discuss bulk fluxes. An exception is Fig. 3.

2) bulk tracer mass fluxes or simply bulk fluxes are the spatially integrated tracer mass fluxes at the boundary between two volumes ($F_{ij}$).

3) total tracer mass transport corresponds to the tracer mass which passes through a boundary during the course of a day ($M_{ij}$).
We see the need to be more explicit with our definitions and have clarified this point in in subsection 2.3. However, we call bulk fluxes often simply 'fluxes' in the text if the context is clear. Stating the full name everywhere would be tiring for the reader. The title of subsection 2.3 has been renamed 'Definition of volumes and fluxes in between' and Sec. 3.2 to 'bulk tracer mass fluxes'.

**28) l. 148: As long as the valley atmosphere is ...**
We rephrased the sentence to:
As long as the valley atmosphere is characterized ...

**29) l. 154: What does more useful mean?**
We change the corresponding sentence to:
'However, it is hard to separate the SWL from the CBL in this region, so that the chosen

definition is more appropriate for quantifying tracer fluxes in and out of the SWL.'

**30) l. 157: What time is sunrise? At initialization time? Please clarify.**
The 'sun rises' – or more specifically heating starts – when the model is initialized, at 06 LT. This information has been added for clarification in section 2.1.

**31) l. 159-160: How is the breakup detected? I believe that once the stable core is eroded the heat flux criterion detects the inversion around ridge height as $h_{cbl}$.**
Exactly. We have added this information in section 2.3.

**32) l. 166: Why is that useful?**
Reviewer 1 asked the very same question. Please refer to the replay to comment 13 of reviewer 1.

**33) l. 174-175: Are the budgets for the different volumes closed? What about sub-grid scale fluxes? Did the authors quantify these?**
Sub-grid scale fluxes are parametrized. For example, the vertical sub-grid scale tracer flux is $w'\rho'_{tr} = -K_v \frac{d\rho_t r}{dz}$, where $K_v$ is the vertical eddy diffusivity. Similar equations hold for the horizontal sub-grid scale fluxes. The sub-grid scale tracer flux is a part of the total flux as defined in Eq. 2.

The tracer mass budgets of the volumes 1-3 are not perfectly closed in the morning since interpolation and changing volume sizes lead to errors. This is especially an issue in the early morning hours when V1 is still small and the fluxes into and out of V1 are large. The error decreases with time and at 9 LT, the budgets are closed.

We have added information about this issue to the manuscript (line. 207 ff).

**34) l. 196: Do the authors have an explanation for the large difference?**
The imbalance of the budgets in volumes 1-3 (see response to previous comment) leads to errors of the calculation of $F_{23}$ in the morning. Since $F_{23}$ is calculated as a residual from the budget equations for V2 and V3 independently, the individual results differ and averaging the two results helps to reduce the error. The difference decreases and reaches zero at 9 LT.

The explanation in the methods-section has been improved (line. 207 ff).

**35) l. 206: What are the similar characteristics?**
We added this information. The revised text reads:
'The daytime SWL has characteristics similar to a CBL over a flat surface such as the valley floor, with a superadiabatic surface layer followed by a well-mixed layer and a capping inversion.'

**36) l. 232: It is not so much a failure but rather results from the criterion itself. 37) l. 234: The heat-flux criterion is not used for the upper boundaries of V2 and V3. The authors should be more precise here. 38) l. 235: How is the boundary between V1 and V2 determined?**

Comments 36, 37 and 38 refer to the same paragraph and we decided to answer these questions together. Regarding the boundary between V1 and V2: As the text says, the boundary between V1 and V2 is $\bar{h}_{\rm cbl}$. A line at this height is extended towards the valley side walls as indicated as a straight line at the height $\bar{h}_{\rm cbl}$ in Fig. 1. $\bar{h}_{\rm cbl}$ is determined as described in sec. 2.3.

The rephrased paragraph reads now:

Since the $w$-criterion cannot yield a suitable CBL height in the center of the valley and the $\theta$-gradient-criterion is sensitive to small temperature gradients, we use the *heat-flux-criterion* in the remainder of the paper for both the CBL and SWL height. Before the breakup of the inversion (regime 1), $\bar{h}_{\rm cbl}$ is used as the upper (lower) boundary of $V_1$ ($V_2$ and $V_3$) while $h_{\rm swl}$ serves as the lateral boundary between $V_2$ and $V_3$ (see Fig. 1). A spatial average using a Hann-window over 2 km is applied to $d_{\rm swl}$ before calculating $h_{\rm swl}$ to smooth wiggles which would be unsuitable for the definition of $V_2$ and its border to $V_3$. Finally, the upper boundary of $V_2$ and $V_3$ is marked by the crest height $h_c$. After the breakup (regime 2), $V_2$ and $V_3$ become zero and the upper boundary of $V_1$ becomes $h_c$.

**39) l. 243: What does in the morning mean? I guess heating starts once the simulations are started? CBL instead of mixed layer.**

Exactly, morning means 06 LT (start of the simulations). We see that the formulation may suggest that the heating starts later. We removed the words 'in the morning' from the sentence in question. Also, mixed layer is replaced by CBL.

**40) l. 244: the description of the SWL is not necessary as it is already introduced. Why is the SWL and CBL not indicated in Figs. 2a, b?**

We removed the redundant description of the SWL. In Figs. 2a, b we show the instantaneous fields and we think the SWL and CBL lines are not appropriate here since they are defined for the average fields.

**41) l. 247: here and thereafter: once the abbreviation HI is defined it should also be used.**

We are using the abbreviation HI now consistently.

***42) l.252: description of HI not necessary anymore.***
This passage was rewritten following a request from reviewer 1. More details regarding the thickness of the SWL and the average along-slope wind speed are given. This way, we provide additional information instead of only a description of the HI.

***43) l. 254: CBL instead of mixed layer.***
'Mixed layer' replaced by CBL

***44) l. 260-263: Did the authors investigate the variability of HI for the different output times? This might be interesting to know.***
Yes, we did. Figure 2 shows the HI at 09 and 12 LT and one can see that the HI grows strongly in time and occupies a larger area. Generally, the HI grows weaker again in the afternoon since the forcing and hence the slope winds decrease in strength. When the breakup is reached in the afternoon, this decrease of the HI's strength is faster since the HI is bound to the existence of a stable layer.

The fact that the HI evolves over time is explained briefly in the manuscript (l 278-280), but we decided to not go into great detail to avoid detraction from the main topic of the investigation.

***45) l. 264-265: How did the authors detect the turbulent mixing?***
Turbulent heat fluxes and the presence of TKE indicate turbulent mixing. We didn't want to add figures of the TKE budget to the paper since this seemed to be distracting from out main goal.

***46) l. 271-274: This rather belongs to the description of the other simulations in the next section.***
We disagree. Although we describe the reference run in this section it is certainly helpful to have at hint of how other runs evolve since we describe bulk fluxes for these simulations in the next section.

***47) l. 267-268: Where does this second and stronger circulation come from? Is this symmetric, i.e. does it occur on the other ridge as well? This circulation is right at the boundary of the model domain. Is this a numerical boundary effect? Why did the authors choose the boundaries of the model domain so close to the valley? Did the authors perform sensitivity runs with***

*other domain boundaries? According to the definition of the authors this circulation occurs in the free atmosphere, which is somewhat contradictory as it obviously is a boundary layer process (see major comment 1).*

This 'second and stronger circulation' is actually the general valley circulation which consists of the slope wind layers and the subsidence in the center of the valley. It is symmetric in our simulations. In the presence of a stable core, a weaker, secondary circulation develops which is associated with the HI. This is not a numerical artefact caused by boundary conditions since it has been found by other authors (i.e., Schmidli, 2013; Wagner et al., 2015a) as well, including in simulations with a topography consisting of a double ridge with adjacent plains. We have chosen to set the boundaries at the mountain peak since this is the minimal domain size required to achieve the goals of this study.

*48) l. 278: into the CBL*
Corrected.

*49) l. 284: If regime 2 is never reached in S1N10 it can't be reached earlier or later in other runs.*
We agree that our statement is misleading. The revised sentence reads:
'Simulations with a stronger forcing or a weaker stability than the reference case S1N10 exhibit a similar evolution, but the inversion breakup (regime 2) is reached. In the case of a weaker forcing or a stronger stability than S1N10, the breakup is never reached.'

*50) l. 290: bulk or volume fluxes? See previous comment.*
The section has been renamed to 'bulk tracer mass fluxes' (see reply to comment 27). Bulk fluxes are already defined at the border between volumes and there is no need to re-iterate.

*51) l. 293: the authors use SWL and V2 synchronous (here and at several locations in the text). It would improve readability if the authors used only one term.*
Although SWL and $V_2$ are identical in the context of this study we use both terms simultaneously to remind the reader which volume is which. The numbered volumes are useful to define bulk fluxes and associated indices and it is more natural to write about $V_2$ in this context.

*52) l. 312: How can the normalized flux exceed 100%? The definition of the top of volume V1 during regime 2 is not clear to me. How deep was the neutrally stratified layer? The authors stated that is exceed the ridge height (line.272). See also major comment 1.*

The normalized fluxes can exceed 100 % because the fluxes are normalized with the constant tracer mass flux at the surface. If tracer mass accumulates in the CBL, a volume of heavily polluted air will eventually be exported via the SWL, or is exported directly by thermals originating from the valley floor. This way, the tracer mass flux is stronger than the source flux, but only for a short period of time. Since we use a geometric definition of the valley volume, the top of V1 is crest height during regime 2, although the neutrally stratified layer exceeds out of the valley.

We have clarified the first point in the manuscript (line 340ff):

Due to this slower increase, more tracer mass accumulates before noon in $V_1$. Eventually, a volume of heavily polluted air will enter the SWL, be advected along the slope and leave the valley at crest height. This leads to peak values for $f_{24}$ of 125 %. This flux decreases again once the particularly tracer-rich air has left the valley.

The fact that the top of V1 during regime 2 is $h_c$ is made more explicit in Sec. 2.3.

*53) l. 338-339: This sentence is irrelevant.*

Sentence removed.

*54) l. 341: How is the total tracer mass calculated? Between 6 and 12 LT or between 6 and 18 LT as it is stated in Fig. 5?*

The statement 'between 6 LT and 12 LT' from line 341 is a typo. It should read between 06 LT and 18 LT, which is a 12 h period.

*55) l. 364: Why does the surface sensible heat flux decrease sharply at 18 LT? I thought it follows a sine curve?*

The wording is indeed misleading. Since the criterion to detect $h_{cbl}$ and $d_{swl}$ depends on a threshold, these heights jump to lower levels as the threshold is met at this lower level. This happens as the surface sensible heat flux approaches zero in the late afternoon. The surface sensible heat flux follows indeed a sine curve. However, the changes made in reply to comment 2 led to the removal of the text passage in question.

*56) l. 374: Dependence on forcing amplitude and initial stability?*

This is indeed a better title for this subsection.

*57) l. 378-379: english: For simulations with different than ???*

This sentence requires a complete rephrasing.

'The overall structure of the valley atmosphere and the magnitudes of tracer fluxes between various volumes are similar when comparing the reference simulations with those with a weaker or stronger forcing.'

**58) l. 381: As far as I understand $F_{12}$ is defined at height $h_{cbl}$ and the horizontal extent of the area reaches from the bottom of the slope ($x = 2$ km) to the slope. With different surface heating I expect the height $h_{cbl}$ and thus the horizontal area to vary strongly. How does this affect the fluxes and what does this mean for the comparability of the different runs? Do the authors consider this in the analysis?**

The reviewer has misunderstood the definition of V2. Its lower boundary is indeed defined at $h_{cbl}$, but its width is given by the horizontal distance between the slope surface and the border between V2 and V3, which is calculated using the $\theta$-gradient criterion. The horizontal area is therefore not as variable as the reviewer might have imagined. Still, it is not constant; correct. The width of the SWL matters indeed for the tracer mass flux (unit g m$^{-2}$ s$^{-1}$), but not for the bulk tracer mass flux (unit g s$^{-1}$), which is integrated over the extend of a border. At the end of the day, we are interested in the amount of tracer which is transported horizontally and vertically and not so much in tracer concentrations at certain locations. Of course, the width of the SWL matters if someone is interested in the tracer concentration over the slope, but this is not the problem we would like to address. We have improved Sec. 2.3 to clarify the definition of V2.

**59) l.390: What is "usual meaning"? I do not understand lower end of the SWL? Please clarify.**

The usual meaning of the other symbols refers to $\rho$ and $c_p$, which are the density of the air and $c_p$, the specific heat capacity at constant pressure, respectively. The lower end of the SWL refers to CBL height. We have clarified this issue (line. 399ff).

**60) l. 393: SWL into the free atmosphere. See major comment 1.**

We replaced the term 'free atmosphere' with 'atmosphere above the valley'. Please refer to our answer to your major comment 1.

**61) l. 423: Change the title of this subsection so that it reflects its content better. E.g. Definition of a dimensionless parameter**

Title of subsection changed to 'A non-dimensional breakup parameter'.

**62) l. 439: English: not further conclusions...**

Spelling corrected.

**63) l. 441: ... important for what?**

For more clarity, we include ' ... important for the total export of tracer mass...' in the sentence.

*64) l. 446-447: .. with strong stability and/or weak forcing ... implies weak stability and/or strong forcing ...*

Sentence rephrased as suggested.

*65) l. 448: As B is defined as the ratio between required and provided energy it does not depend directly depend on the breakup time. It is rather a measure of total mixing. Only its sign gives some information if the breakup is reached (B<1) or not. Thus, I do not understand why the authors call it breakup parameter. In my opinion, this is misleading. Something like "mixing parameter" or "export parameter" would be more appropriate.*

It seems that the reviewer has misunderstood the meaning of the breakup parameter. $Q_{req}$ is the required energy to remove the inversion at sunrise (6 LT). If enough energy will be provided over the course of the day, the breakup will be achieved at some point. The more energy is provided, the earlier the breakup. So, for $B < 1$, this parameter is indeed correlated with the breakup time, hence its name (l 450ff). What the reviewer appears to have in mind would be a parameter $M$ which would be defined as the *currently* required energy to remove the inversion over the total energy provided (or the energy which will be provided until sunset). Such a parameter $M$ could indeed be called a mixing parameter, and it might be interesting as well, but this is not the parameter we had in mind. We do believe that our explanation of the concept is clear enough, so we did not change the text.

*66) l. 459ff: The discussion section should be improved. At the moment it is rather long and repeats a lot of previous studies (e.g. l. 495-503 or l. 523-526) but often misses a critical discussion of the findings of this manuscript to other studies (e.g. l. 489-494).*

We have revised the discussion section thoroughly. Repetitions of previous studies are limited to the minimal information required and critical findings and simplifications are now discussed in more detail.

*67) l. 464: What is inhomogeneous? The terrain or the atmosphere?*

Both. Of course, the spatially inhomogeneous atmosphere is (close to the ground) to a large degree a result of spatially inhomogeneous terrain and land surface features. As part of the general revision of the discussion section (see reply to comment 66), we removed this sentence.

*68) l. 463-488: The sensitivity of the tracer fluxes to the different definitions of the volumes is very interesting. The discussion would benefit from an additional figure visualizing the sensitivity.*

A new figure comparing the fluxes of S1N10 for the heat-flux and the $\theta$-gradient criterion has been added to the manuscript. The paragraph describing this figure and the shown sensitivities has been modified.

*69) l. 495-503: this is pretty much a summary of previous studies. It would be more interesting to discuss the aspects related to terrain geometry (depth, width, slope angles, ...) which have not been considered in this study and maybe speculate about their impact.*

We agree and have improved this part of the discussion. Please refer to our reply to comment no. 66.

*70) l. 553: referring to V2 and V3 in the summary is not ideal, as a reader only reading the summary would not understand it.*

A very good point. We replaced V2 and V3 by the terms 'slope wind layer' and 'stable core'.

*71) l. 555-558: Some more details on the how?*

More details are given in the rephrased bullet point:
The vertical transport of tracer mass from the CBL into the slope wind layer primarily depends on the forcing amplitude. A stronger forcing leads to an earlier onset of the associated tracer mass flux and a sharper increase in time. The horizontal flux from the slope wind layer into the stable core and the export at crest height depend on both forcing amplitude and initial stability. The export decreases drastically with increasing stability and decreasing forcing while the horizontal transport increases.

*72) l. 570-571: Testing the dependence on realistic atmospheric stratifications (i.e. several elevated inversions) would be interesting too.*

We agree. This is added to the list of open questions for future studies.

*73) Fig. 2: How long are the time averages? Are the wind vectors time averaged as well?*

All variables in Fig. 2 c-f are averaged along the $y$-axis and over 41 minutes. Wind vectors are drawn using averaged wind components. This is now indicated in the caption.

*74) Fig. 4: Are the shown fluxes averaged in time or instantaneous? It would be helpful to include the total export into V4 ($f_{24} + f_{34}$).*

Yes, these are averaged fluxes and they consist of mean, resolved and subgrid-scale tracer mass fluxes (see Sec. 2.2). $F_{ij}$ is always the sum of these three terms. A line showing the total export has been added to Fig. 4.

**75) Fig. 8: ".... Of the forcing amplitude, $A_{shf}$, ..."**
Sentence rephrased as suggested.

**References**

Catalano, F., and C.-H. Moeng, 2010: Large-Eddy Simulation of the Daytime Boundary Layer in an Idealized Valley Using the Weather Research and Forecasting Numerical Model. *Bound-Lay. Meteorol.*, **137 (1)**, 49–75, doi:10.1007/s10546-010-9518-8.

Lang, M. N., A. Gohm, and J. S. Wagner, 2015: The impact of embedded valleys on daytime pollution transport over a mountain range. *Atmos. Chem. Phys.*, **15 (20)**, 11 981–11 998, doi:10.5194/acp-15-11981-2015.

Leukauf, D., A. Gohm, M. W. Rotach, and J. S. Wagner, 2015: The Impact of the Temperature Inversion Breakup on the Exchange of Heat and Mass in an Idealized Valley: Sensitivity to the Radiative Forcing. *J. Appl. Meteor. Climatol.*, **54 (11)**, 2199–2216, doi:10.1175/JAMC-D-15-0091.1.

Rotach, M. W., M. Andretta, P. Calanca, A. P. Weigel, and A. Weiss, 2007: Boundary layer characteristics and turbulent exchange mechanisms in highly complex terrain. *Acta Geophys.*, **56 (1)**, 194–219, doi:10.2478/s11600-007-0043-1.

Rotach, M. W., and D. Zardi, 2007: On the boundary-layer structure over highly complex terrain: Key findings from MAP. *Q.J.R. Meteorol. Soc.*, **133 (625)**, 937–948, doi:10.1002/qj.71.

Schmidli, J., 2013: Daytime Heat Transfer Processes over Mountainous Terrain. *Journal of the Atmospheric Sciences*, **70 (12)**, 4041–4066, doi:10.1175/JAS-D-13-083.1.

Schmidli, J., and R. Rotunno, 2010: Mechanisms of Along-Valley Winds and Heat Exchange over Mountainous Terrain. *J. Atmos. Sci.*, **67 (9)**, 3033–3047, doi:10.1175/2010JAS3473.1.

Serafin, S., and D. Zardi, 2010: Daytime Heat Transfer Processes Related to Slope Flows and Turbulent Convection in an Idealized Mountain Valley. *J. Atmos. Sci.*, **67 (11)**, 3739–3756, doi:10.1175/2010JAS3428.1.

Wagner, J. S., A. Gohm, and M. W. Rotach, 2014: The Impact of Horizontal Model Grid Resolution on the Boundary Layer Structure over an Idealized Valley. *Mon. Wea. Rev.*, **142 (9)**, 3446–3465, doi:10.1175/MWR-D-14-00002.1.

Wagner, J. S., A. Gohm, and M. W. Rotach, 2015a: The impact of valley geometry on daytime thermally driven flows and vertical transport processes. *Q.J.R. Meteorol. Soc.*, **141 (690)**, 1780–1794, doi:10.1002/qj.2481.

Wagner, J. S., A. Gohm, and M. W. Rotach, 2015b: Influence of along-valley terrain heterogeneity on exchange processes over idealized valleys. *Atmos. Chem. Phys.*, **15 (12)**, 6589–6603, doi:10.5194/acp-15-6589-2015.

Weigel, A. P., F. K. Chow, and M. W. Rotach, 2007: The effect of mountainous topography on moisture exchange between the "surface" and the free atmosphere. *Boundary-Layer Meteorology*, **125 (2)**, 227–244, doi:10.1007/s10546-006-9120-2.

---

## Author Comment (AC2) · 26 Aug 2016

[revised manuscript text omitted]
                      | 375                            | 6, 10, 14, 18                |

---

## Author Response (AR2)

**Response to the final comments by the Co-Editior Dr. Bernhard Vogel (acp-2016-350)**

Daniel Leukauf, Alexander Gohm and Mathias W. Rotach

September 26, 2016

*1) The abstract is still too long. The finding should be the main part of the abstract, not the method. Please shorten.*

The first paragraph of the abstract has been shortened. The second one remains unchanged.

*2) Page 6, line 163: What do you mean by reliable averages? 'robust averages' might be the better wording.*

We agree that 'robust averages' is the better wording and have changed it accordingly (line 162).

*3) Page 6, line 164: 'a non-dimensional mixing ratio r'. In my view r is by definition the mass mixing ratio.*

Mass mixing ratio is indeed better. We changed the wording as suggested (line 163).

*4) Use 'Brunt-Väisälä frequency' instead of 'buoyancy frequency' throughout the whole text.*

All instances of 'buoyancy frequency' have been replaced by 'Brunt-Väisälä frequency' (lines 7, 29, 140, 147, 269, 570, page 17 footnote and Fig. 8 (caption)).

*5) I see no reason to use the abbreviation SWL and V2 for the same quantity. Moreover it is even confusing.*

We agree that the usage of the abbreviations SWL and V2 at the same time is suboptimal. The difficulty is that the term 'slope wind layer' is used in a slightly more colloquial way (i.e., without a strict definition, since multiple definitions exist) in the introduction, but the well defined volume (V2) is introduced later on in the methods section. The numbering of the volumes (V1, V2, V3) is also necessary and convenient since the fluxes $f_ij$ are named using the corresponding indices. In other words, V2 is the slope wind layer after one particular definition and hence the possible confusion.

We have decided to solve this problem by removing the abbreviation SWL and replace it with 'slope wind layer' in the introduction, the discussion, the methods and the general description of the flow. In these section, a more intuitive name seems more appropriate and helpful to understand the presented research. In the sections 3.2, 3.3 and 3.4, where the results are presented with reference to related fluxes, we have replaced SWL by V2. Also, some occasions of 'slope wind layer depth' have been changed to '$d_{swl}$'. These changes affect the lines 31, 43, 44, 45, 48, 51, 53, 57, 68, 75, 92, 104, 172, 173, 174, 175, 178, 183, 225, 227, 230, 233, 234, 244, 245, 249, 256, 257, 260, 270, 274, 276, 277, 280, 286, 287, 291, 295, 296, 306, 307, 310, 318, 319, 323, 325, 326, 332, 345, 351, 354, 362, 371, 394, 396, 397, 401, 402, 407, 412, 415, 417, 420, 421, 422, 423, 424, 426, 480, 482, 483, 484, 487, 491, 492, 493, 498, 500, 502, 503 and 528.